# Long-term antigen exposure irreversibly modifies metabolic requirements for T cell function

Marie Bettonville[1†], Stefania d'Aria[1†], Kathleen Weatherly[1], Paolo E Porporato[2‡], Jinyu Zhang[1§], Sabrina Bousbata[3], Pierre Sonveaux[2], Michel Y Braun[1*]

[1]Institute for Medical Immunology, Université Libre de Bruxelles, Gosselies, Belgium; [2]Pole of Pharmacology & Therapeutics, Institut de Recherche Expérimentale et Clinique, Université Catholique de Louvain, Brussels, Belgium; [3]Laboratory of Molecular Parasitology, Proteomic Platform, Institute of Molecular Biology and Medicine, Université Libre de Bruxelles, Gosselies, Belgium

*For correspondence:
mbraun@ulb.ac.be

[†]These authors contributed equally to this work

Present address: [‡]Department of Molecular Biotechnology and Health Sciences, University of Turin, Turin, Italy; [§]Department of Clinical Microbiology and Immunology, Third MilitaryMedical University, Chongqing, China

Competing interests: The authors declare that no competing interests exist.

**Abstract** Energy metabolism is essential for T cell function. However, how persistent antigenic stimulation affects T cell metabolism is unknown. Here, we report that long-term in vivo antigenic exposure induced a specific deficit in numerous metabolic enzymes. Accordingly, T cells exhibited low basal glycolytic flux and limited respiratory capacity. Strikingly, blockade of inhibitory receptor PD-1 stimulated the production of IFNγ in chronic T cells, but failed to shift their metabolism towards aerobic glycolysis, as observed in effector T cells. Instead, chronic T cells appeared to rely on oxidative phosphorylation (OXPHOS) and fatty acid oxidation (FAO) to produce ATP for IFNγ synthesis. Check-point blockade, however, increased mitochondrial production of superoxide and reduced viability and effector function. Thus, in the absence of a glycolytic switch, PD-1-mediated inhibition appears essential for limiting oxidative metabolism linked to effector function in chronic T cells, thereby promoting survival and functional fitness.
DOI: https://doi.org/10.7554/eLife.30938.001

## Introduction

Quiescent T cells rely on oxidative phosphorylation (OXPHOS) for energy production (*Plas et al., 2002*). However, once activated by foreign antigens, they shift from an energetically efficient oxidative metabolism to a highly glycolytic and glutamine-dependent metabolic program (*Chang et al., 2013*; *Nakaya et al., 2014*; *Gubser et al., 2013*; *van der Windt et al., 2013*). This metabolic switch endows T cells with the capacity to direct rapidly metabolic intermediates toward the synthesis of macromolecules required for proliferation and effector function. Interestingly, when aerobic glycolysis is impaired, T cells can also rely on OXPHOS alone to become activated and proliferate (*Chang et al., 2013*; *Sena et al., 2013*). In this case, T cell effector function is compromised, with impaired cytokine production caused by posttranscriptional regulation of glycolytic enzymes (*Chang et al., 2013*). Thus, in addition to providing precursors for biomass production in activated T cells, aerobic glycolysis allows for the acquisition of full effector functions. The capacity of T cells to limit glycolysis has been linked to the expression of the cell surface inhibitory receptor PD-1 (2), which is increased following T cell activation (*Agata et al., 1996*). The function of inhibitory receptors such as PD-1 is to attenuate signaling downstream of the TcR, leading to suppression of T-cell proliferation, cytokine production and cytolytic function (*Yokosuka et al., 2012*). Recently, PD-1 signaling was shown to regulate T cell function by inhibiting glycolysis (*Patsoukis et al., 2015*; *Bengsch et al., 2016*), and to promote T cell survival through lipolysis and OXPHOS fueled by fatty acids (*Patsoukis et al., 2015*).

Severe combined immunodeficiency disease (SCID) causes patients to have severe defects in the function of their lymphocytes (*Fischer et al., 2015*). Most cases of SCID are due to mutations in the gene encoding common gamma chain (γc), a protein shared by several receptors for interleukins IL-2, IL-4, IL-7, IL-9, IL-15 and IL-21 (*Malek et al., 1999*). Because these cytokines are needed for T cell development, mutations resulting in expression defects cause complete absence of adaptive immunity (*Fischer et al., 2015*). Currently, allogeneic hematopoietic stem cell (HSC) transplantation is the only available treatment for SCID. However, recipient-specific donor T cells present in the graft can cause significant inflammatory events in allogeneic HSC-transplanted SCID patients, including persistent chronic graft-versus-host disease (GVHD) (*Neven et al., 2009*). In experimental animals transplanted with HSC after myeloablative treatment, PD-1 was shown to contribute to the suppression of T cell-mediated GVHD, and PD-1 blockade was associated with an increased glycolytic metabolism in GVHD T cells (*Saha et al., 2013*; *Fujiwara et al., 2014*). Whether PD-1-mediated regulation of T cell metabolism can control inflammation in transplanted SCID patients is unknown. In this context, our study aimed to investigate the metabolic requirements needed for the production of pro-inflammatory cytokine gamma interferon (IFNγ) in CD4$^+$ T cells under persistent exposure to recipient's alloantigens. We show here that, unlike alloreactive early effector T cells in which the production of IFNγ requires a metabolic switch towards glycolysis, long-term chronically stimulated CD4$^+$ T cells (chronic T cells) can mount a rapid effector response despite their apparent failure to undergo a glycolytic switch. Instead, chronic T cells appeared to rely mostly on OXPHOS to support their function. More strikingly, although chronic CD4$^+$ T cells do express PD-1 and PD-1 activation repressed their function, we report here that the inability of these cells to increase their glycolytic rate after activation is independent of PD-1 signaling.

## Results

### Persistent exposure to alloantigens modifies the metabolic requirements for effector function in alloreactive CD4$^+$ T cells

To address the molecular mechanisms regulating the metabolism of alloreactive T cells in SCID recipients, BALB/c (H-2$^d$) CD4$^+$ T cells were transferred into allogeneic *Rag2$^{-/-}$ Il2rγ$^{-/-}$* B6 (H-2$^b$). Though the relative number of donor T cells present in the recipient's spleen increased with time, their absolute numbers were similar at 6 to 21 days post-transfer (*Figure 1A*). After transfer, most T cells were activated, as indicated by their expression pattern of CD44 and CD62L, and expressed PD-1 (*Figure 1B*). In vitro stimulation with irradiated allogeneic cells stimulated the production of IFNγ in both T cell populations (*Figure 1C*), and blockade of PD-1/PD-L1 interaction significantly increased the amount of cytokine produced, confirming that T cell-mediated GVHD is regulated by PD-1 (*Figure 1C*). We noticed, however, that T cells produced higher amounts of IFNγ at day seven than after day 21 (*Figure 1C*). Considering that glycolysis has been largely accounted for the effector function of activated T cells (*Chang et al., 2013*; *Gubser et al., 2013*; *van der Windt et al., 2013*), whether a lower metabolism could limit IFNγ production in day 21 T cells was investigated. Inhibiting glycolysis with glucose analog 2-deoxy-D-glucose (2-DG) is known to inhibit T cell function (*Woodward and Hudson, 1954*). Day 6 T cells were highly sensitive to glycolysis inhibition, whereas day 21 T cells were not inhibited (*Figure 1D*). Taken together, these observations led to the hypothesis that chronic antigenic stimulation could have modified the metabolic requirements for effector function and that reduced IFNγ production would be a direct consequence of limited metabolism in chronic alloreactive T cells.

### Persistent exposure to minor histocompatibility antigens modifies the metabolism of CD4$^+$ T cells

To be able to identify T cells whose activity was specifically modulated by chronic antigen stimulation, we transferred monoclonal anti-male CD4$^+$ TcR-transgenic T cells into male *Rag2$^{-/-}$ Il2rγ$^{-/-}$* B6 recipients. After transfer, most T cells were activated, as indicated by their expression pattern of CD44 and CD62L (*Figure 2A*). Long-term antigen exposure (21 days) up-regulated PD-1 expression on male-specific CD4$^+$ T cells (*Figure 2A*). Spleen cells isolated from adoptively transferred recipients, which contained male-specific T cells and male APCs, produced IFNγ after PD-1/PD-L1 blockade (*Figure 2B*). Cytokine production was the result of T cell activity since IFNγ was not detected

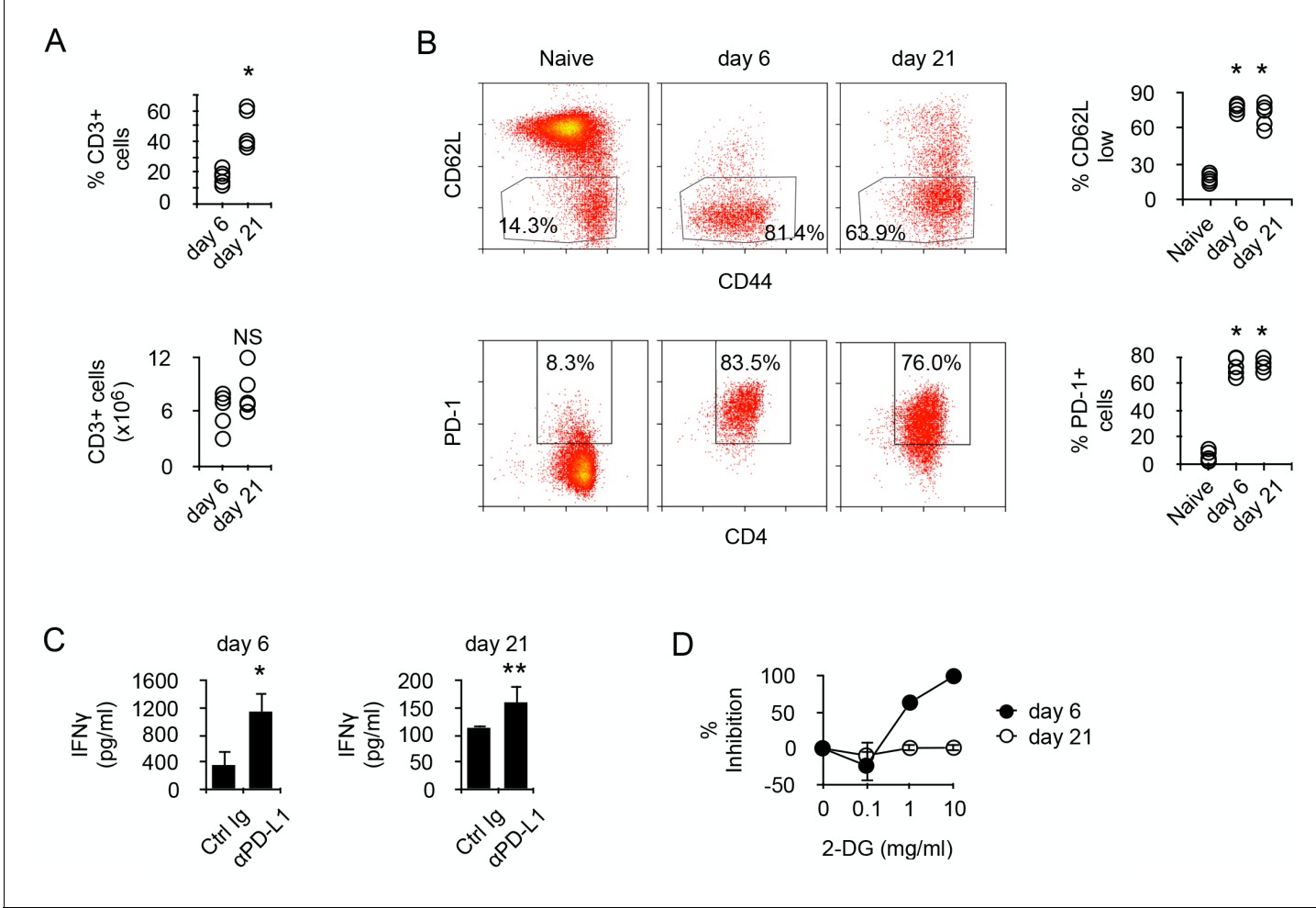

**Figure 1.** Prolonged chronic alloantigen stimulation alters functional metabolic requirements in alloreactive CD4 +T cells. (A) Frequencies and numbers of CD3[+] T cells in the spleen of non-irradiated B6 (H-2b) *Rag2[-/-] Il2rg[-/-]* recipients 6 and 21 days after reconstitution with purified BALB/c (H-2d) CD4[+] T cells. Represented data are means ± SEM. Data presented are representative of two independent experiments with 4–5 mice in each experimental group. *indicates p=0.0079 by the Mann-Whitney test. NS indicates non-significant. (B) Phenotype of chronic alloreactive CD4[+] T cells. Spleen CD3[+] cells from mice as described in (A) were analyzed for their expression of CD44, CD62L and PD-1 by flow cytometry. Represented data are means ± SEM. Data presented are representative of two independent experiments with 4–5 mice in each experimental group. *indicates p=0.0079 (compared to naive) by the Mann-Whitney test. (C) IFNγ production by alloreactive CD4[+] T cells purified from mice as described in (A) and stimulated by irradiated B6 splenocytes in the presence of control or neutralizing anti-PD-L1 antibodies. Represented data are means ± SEM of five replicates and are representative of 2 independent experiments. * indicates p=0.0079 and ** indicate p<0.0286 by the Mann-Whitney test. (D) Inhibition (%) of IFNγ production by alloreactive CD4[+] T cells purified and stimulated as in (C) with different doses of 2-Deoxy-D-glucose (2-DG). Represented data are means ± SEM of five replicates and are representative of 2 independent experiments.

DOI: https://doi.org/10.7554/eLife.30938.002

after blocking T cell antigen recognition (*Figure 2B*). The capacity of PD-1 to inhibit T cell activity was further documented by the observation that engagement of PD-1 by immobilized PD-L1-Ig prevented IFNγ production in anti-CD3/CD28-stimulated purified chronic CD4[+] T cells (*Figure 2C*). Moreover, blockade of PD-1/PD-L1 interaction in an antigen-specific assay decreased by one log the amount of male peptides required for T cell activation and IFNγ production (*Figure 2D*). Taken together, these results demonstrated that PD-1 is responsible for limiting the functional responsiveness of chronic anti-male CD4[+] T cells, and that this regulation is carried out by raising their TcR activation threshold. Moreover, injecting anti-PD-L1 antibodies to male recipients increased sharply the number of T cells infiltrating several organs, such as liver, gut, skin and kidney (*Figure 2E* and data not shown), demonstrating that PD-1 also controlled in vivo the activity of chronic CD4[+] T cells. Importantly, when stimulated in vitro by anti-CD3/CD28 antibodies, purified chronic T cells were

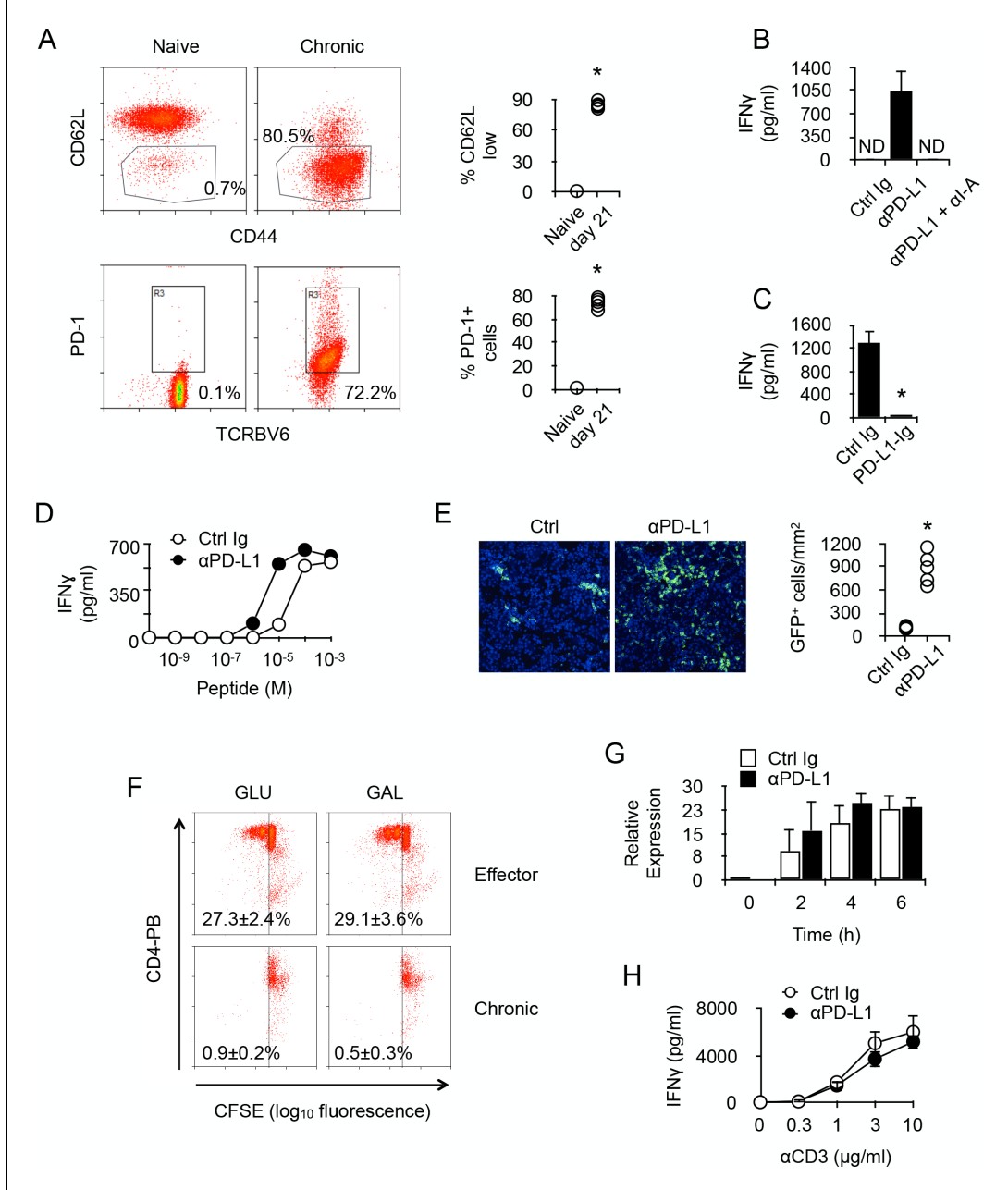

**Figure 2.** PD-1 controls the activity of chronic anti-male CD4 +T cells. (**A**) Phenotype of chronic anti-male CD4[+] T cells. Spleen CD3[+] cells from normal Marilyn mice (naive) or from non-irradiated B6 male (H-2b) *Rag2[-/-] Il2rg[-/-]* recipients 21 days after reconstitution with purified Marylin CD4[+] TCRBV6[+] T cells (chronic), were analyzed for their expression of CD44, CD62L and PD-1 by flow cytometry. Represented data are means ± SEM. Data presented are representative of two independent experiments with 4–5 mice in each experimental group. * indicates p=0.0079 (compared to naive) by the Mann-Whitney test. (**B**) IFNγ production in the culture supernatant of spleen cells isolated from non-irradiated male B6 (H-2b) *Rag2[-/-] Il2rg[-/-]* recipients 21 days after reconstitution with purified Marylin CD4[+] TCRBV6[+] T cells. Control, anti-PD-L1 and/or anti-I-A/I-E neutralizing antibodies were added to the cultures. Represented data are means ± SEM of five replicates. Data presented are representative of three independent experiments. ND indicates below detection limits. (**C**) IFNγ production by CD4[+] T cells purified from non-irradiated B6 male (H-2b) *Rag2[-/-] Il2rg[-/-]* recipients 21 days after reconstitution with purified Marylin CD4[+] TCRBV6[+] T cells. Cells were stimulated with anti-CD3/CD28 antibodies in the presence of immobilized Ctrl Ig or PD-L1-Ig fusion proteins. Represented data are means ± SEM of five replicates. Data presented are representative of three independent experiments. * indicates p=0.0079 (compared to Ctrl) by the Mann-Whitney test. (**D**) IFNγ production by CD4[+] T cells purified as in C and stimulated with male peptide-loaded B6 *Cd3[-/-]* bone marrow-derived dendritic cells in the presence of Ctrl or anti-PD-L1 neutralizing antibodies. Represented data are means ± SEM of three replicates. Data presented are representative of two independent experiments. (**E**) Cell counts of GFP[+] T cells infiltrating the liver of non-irradiated male B6 (H-2b) *Rag2[-/-] Il2rg[-/-]* recipients 21 days after reconstitution with purified Marylin GFP[+] CD4[+] TCRBV6[+] T cells and that received i.p. Ctrl or anti-PD-L1 neutralizing antibodies. * indicates p=0.0079 (compared to Ctrl antibody group) by the Mann-Whitney test. Represented

*Figure 2 continued on next page*

*Figure 2 continued*

data are means ± SEM. Data presented are representative of two independent experiments with 4–5 mice in each experimental group. (F) In vitro proliferation is assessed by fluorescent decay in CFSE-stained chronic or effector T cells after anti-CD3/CD28 stimulation. (G) IFNγ mRNA relative expression by anti-CD3/CD28-stimulated CD4$^+$ T cells purified from non-irradiated B6 male (H-2b) *Rag2$^{-/-}$ Il2rg$^{-/-}$* recipients 21 days after reconstitution with purified Marylin CD4$^+$ TCRBV6$^+$ T cells. Purified T cells were stimulated in the presence of anti-PD-L1 or control antibodies for 2, 4 and 6 hr. No significant difference was found between the two conditions. (H) IFNγ production by anti-CD3/CD28-stimulated CD4$^+$ T cells purified from non-irradiated B6 male (H-2b) *Rag2$^{-/-}$ Il2rg$^{-/-}$* recipients 21 days after reconstitution with purified Marylin CD4$^+$ TCRBV6$^+$ T cells. Purified T cells were stimulated in the presence of anti-PD-L1 or control antibodies. No significant difference was found between the two conditions.

DOI: https://doi.org/10.7554/eLife.30938.003

unable to proliferate (*Figure 2F*), suggesting that increased organ infiltration by T cells observed after anti-PD-L1 antibody treatment was not due to T cell proliferation but enhanced T cell recruitment.

Given that PD-L1 has been identified as a potent ligand for molecules other than PD-1 and direct interaction with CD80 has proven to be inhibitory during cell to cell contacts between neighboring T cells (*Butte et al., 2007*), there was the possibility that our observations in vitro, since we used purified T cells, could result from T-T cell contacts involving PD-L1 and CD80. To investigate this, we stimulated with anti-CD3/CD28 antibodies purified chronic CD4$^+$ T cells in the presence of anti-PD-L1 antibodies. As seen in *Figure 2G and H*, adding PD-L1-specific antibodies did not improve T cells' capacity to produce IFNγ after activation. These results demonstrated that, in our in vitro experiments, PD-L1/CD80 interactions did not regulate the activity of chronic T cells.

In order to characterize the metabolic activity of chronically-stimulated CD4$^+$ T cells, anti-male CD4$^+$ T cells were analyzed for their cytoplasmic protein content by 2D-gel electrophoresis before and after chronic antigen exposure (*Supplementary file 1*). As displayed in *Table 1*, long term

**Table 1.** Proteins down-regulated in chronic T cells.

| Pathway | Gene symbol | Protein name | Fold | P value |
|---|---|---|---|---|
| Glycolysis | *Pfkp* | Phosphofructokinase | 0.76 | 0.0320 |
| | *Gapdh* | Glyceraldehyde-3-phosphate dehydrogenase | 0.68 | 0.0150 |
| | *Pgk1* | Phosphoglycerate kinase 1 | 0.62 | 0.0020 |
| | *Pkm* | Pyruvate kinase | 0.64 | 0.0090 |
| Non-oxidative Pentose Phosphate | *Taldo1* | Transaldolase 1 | 0.68 | 0.0030 |
| | *Tkt* | Transketolase | 0.52 | 0.0140 |
| Citric Acid Cycle | *Got2* | Glutamatic-oxaloacetic transaminase 2, mitochondrial | 0.58 | 0.0030 |
| | *Aco2* | Aconitase 2, mitochondrial | 0.54 | 0.0200 |
| | *Mdh2* | Malate dehydrogenase 2,NAD (mitochondrial) | 0.36 | 0.0010 |
| Fatty Acid β-oxidation | *Acad1* | Acyl-Coenzyme A dehydrogenase | 0.63 | 0.0090 |
| | *Hadha* | Hydroxyacyl-Coenzyme A dehydrogenase/ 3-ketoacyl-Coenzyme A thiolase/enoyl-Coenzyme A hydratase (trifunctional protein), alpha subunit | 0.47 | 0.0110 |
| Respiratory Chain | *Uqcrc1* | Ubiquinol-cytochrome c reductase core protein 1 | 0.80 | 0.0130 |
| | *Atp5h* | ATP synthase, H + transporting, mitochondrial F0 complex, subunit D | 0.64 | 0.0460 |
| Glutaminolysis | *Tgm1* | Transglutaminase 1, K polypeptide | 0.40 | 0.0040 |
| Protection against oxydative stress | *Prdx2* | Peroxiredoxin 2 | 0.67 | 0.0310 |
| | *Txn1* | Thioredoxin 1 | 0.52 | 0.0100 |
| | *Prdx1* | Peroxiredoxin 1 | 0.43 | 0.0004 |
| | *Park7* | Parkinson disease (autosomal recessive, early onset) 7 | 0.42 | 0.0030 |

DOI: https://doi.org/10.7554/eLife.30938.005

antigen exposure down-regulated the expression of several enzymes involved in cellular metabolism. Glycolysis was the most affected metabolic pathway with the apparent down-regulation of phospho-fructokinase (PFKp), glyceraldehyde-3-phosphate dehydrogenase (GAPDH), phosphoglycerate kinase (PGK1) and pyruvate kinase (PKM). Two major enzymes involved in the non-oxidative phase of the pentose phosphate pathway, transaldolase (TALDO1) and transketolase (TKT), were also found down-regulated after chronic antigen exposure. The citric acid cycle and the lipid oxidation pathway were also affected, since aconitate hydratase (ACO2), glutamate oxaloacetate transaminase 2 (GOT2), malate dehydrogenase 2 (MDH2), 3-hydroxyacyl-CoA dehydrogenase (HADH) and long-chain specific acyl-CoA dehydrogenase (ACAD1) expression decreased significantly in chronic CD4$^+$ T cells. These findings suggested that chronic antigenic stimulation resulted in a general metabolic deficit in male-specific CD4$^+$ T cells. To further explore this possibility, chronic anti-male CD4$^+$ T cells were purified and compared for their metabolic activity with naive T cells or effector T cells, obtained by immunization with irradiated T cell-deficient *Cd3$^{-/-}$* male spleen cells. Basal glycolytic flux, assessed by measuring ECAR, was high in control effector T cells, but similarly low in naive and chronic CD4$^+$ T cells (*Figure 3A*). In the same line, basal respiration was significantly higher in effector T cells compared to naive and chronic T cells (*Figure 3A*). A significant spare respiratory capacity (SRC) was nevertheless detected in chronic T cells after mitochondrial uncoupling (*Figure 3A*). However, compared with naive or effector T cells, it was modest and was correlated with a lower mitochondrial mass (*Figure 3B*) and a reduced capacity to synthesize ATP after activation (*Figure 3C*). Taken together, these results demonstrated that chronic CD4$^+$ T cells have a low energy metabolism and a reduced capacity to produce cellular energy.

## Effector function of chronic CD4$^+$ T cells does not require a metabolic shift towards glycolysis

We then investigated whether the production of IFNγ by chronic T cells requires a glycolytic switch. Chronic CD4$^+$ T cells were sorted and compared with purified naive T cells or effector T cells. As expected, following activation, naive T cells secreted large amount of IL-2 but, in the absence of the polarizing cytokine IL-12, they failed to produce IFNγ. After activation, effector T cells produced large quantities of both IL-2 and IFNγ (*Figure 3D*). In vitro activation stimulated glycolysis in naive and effector T cells, as evidenced by the time-dependent production of lactate in the culture medium, which inversely correlated with glucose consumption (*Figure 3D*). On the contrary, very low levels of lactate and no evidence of glucose consumption were detected for chronic T cells after activation, suggesting that glycolysis was turned down in these cells (*Figure 3D*). Remarkably, lack of glucose consumption did not prevent chronic T cells to produce IFNγ following activation (*Figure 3D*). Thus, chronic exposure of CD4$^+$ T cells to recipient's antigen untied IFNγ production from the need to undergo a glycolytic switch. Accordingly, unlike in effector T cells where glycolytic inhibitor 2-DG strongly repressed IFNγ production, 2-DG only weakly inhibited IFNγ production by chronic T cells (*Figure 4A*). Taken together, these results revealed that glycolysis is deficient in chronic CD4$^+$ T cells and does not support effector function. For glucose uptake, lymphocytes primarily rely on glucose transporter GLUT1, and T cell activation stimulates GLUT1 expression (*Rathmell et al., 2000*). GLUT1 up-regulation that followed anti-CD3/CD28 activation in effector T cells was not observed in chronic T cells (*Figure 3F*). Accordingly, the uptake of fluorescent glucose analog 2-(N-(7-nitrobenz-2-oxa-1,3-diazol-4-yl)amino)−2-deoxyglucose (2-NBDG) (*Yoshioka et al., 1996*) was lower in chronic T cells after anti-CD3/CD28-mediated stimulation (*Figure 3G*). Thus, defective glycolysis in chronic CD4$^+$ T cells could result at least in part from limited glucose entry into the cells.

Since Wherry and co-workers have recently demonstrated that restoring function in virus-specific exhausted CD8$^+$ T cells strictly relies on glycolysis (*Bengsch et al., 2016*), we hypothesized that chronic T cells in the present study might have reached a deeper state of exhaustion where glycolysis was not anymore available for T cell function. To investigate this possibility, we analyzed chronic T cells during the first week of persisting male antigen stimulation, before severe dysfunction develops. As depicted in *Figure 3H*, day seven chronic T cells exhibited a substantial glycolytic capacity whereas, as expected, day 21 chronic T cells did not (*Figure 3H*). Contrary to what was observed in day 21 T cells, the capacity to produce IFNγ in vitro after anti-CD3/CD28 stimulation was very sensitive to glucose metabolism inhibition by 2-DG in day 7 T cells (*Figure 3I*). Moreover, day 7 T cells also expressed lower levels of PD-1, LAG3 and 2B4 than day 21 T cells (*Figure 3J*). These

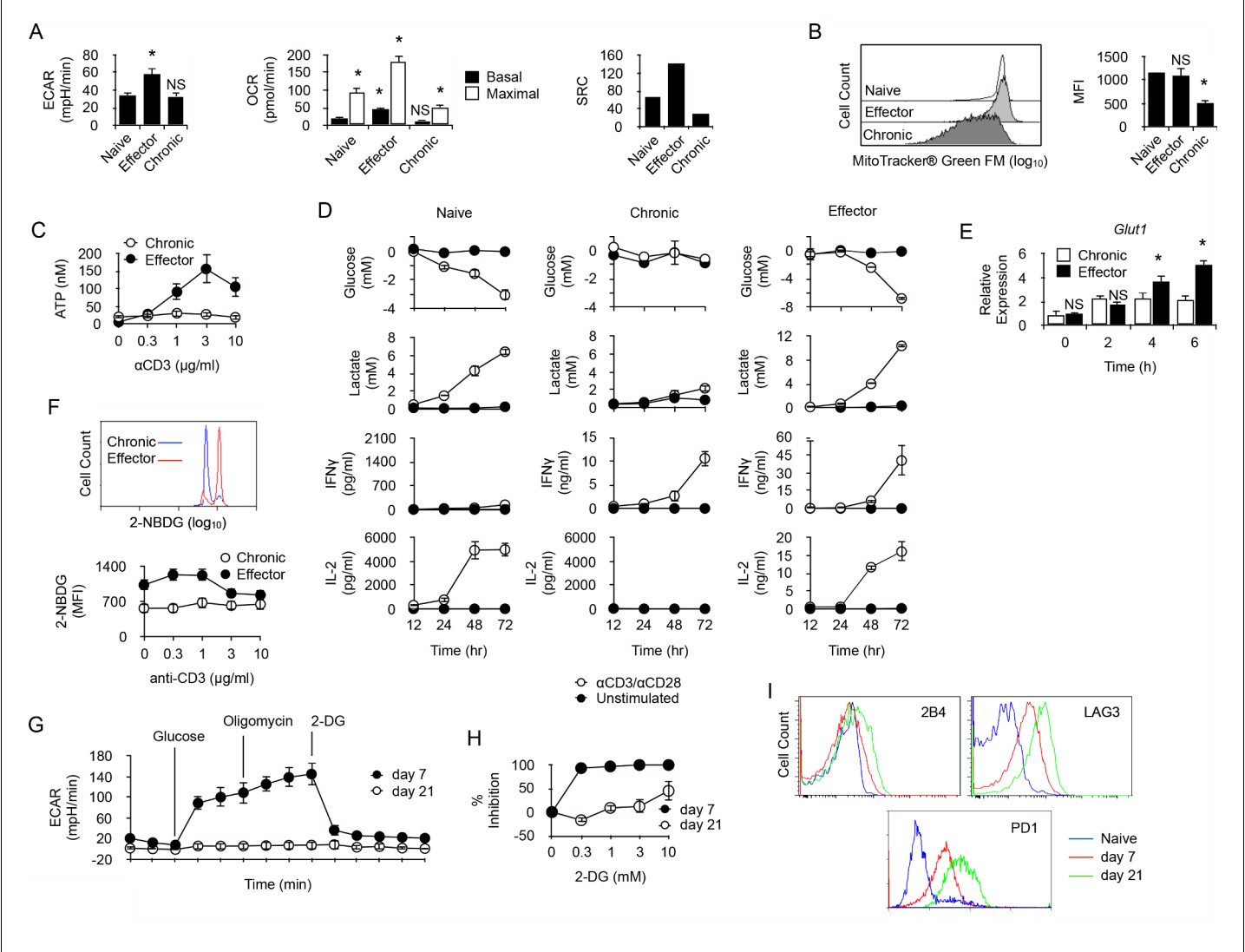

**Figure 3.** Long-term chronic anti-male CD4+T cells exhibit a low metabolism and are unable to shift their metabolism to produce IFNγ after activation. (A) Basal ECAR (left) and basal and maximal OCR (right) of naive, effector or chronic CD4+ T cells. Data (means ± SEM of five replicated cultures) are representative of two independent experiments. * indicates p=0.0079 by the Mann-Whitney test. (B) Mitochondria mass of effector or chronic CD4+ T cells labeled with MitoTracker Green FM. Means ± SEM of MFI from T cells isolated from 4 mice per group is shown. Data are representative of two independent experiments. * indicates p=0.0079 by the Mann-Whitney test. NS is non-significant. (C) ATP in effector or chronic CD4+ T cells before and after anti-CD3/CD28 antibody stimulation. Data (means ± SEM of five replicated cultures) are representative of two independent experiments. (D) Purified naive, effector or chronic CD4+ T cells were stimulated with anti-CD3/CD28 antibodies or left unstimulated. After 12, 24, 48 and 72 hr, culture supernatants were harvested and glucose, lactate, IL-2 and IFNγ concentrations were determined as described in Material and methods. Data (means ± SEM from five replicated cultures) are representative of two independent experiments. (E) GLUT1 mRNA expression was assessed by quantitative PCR in effector and chronic CD4+ T cells after activation with PMA/ionomycin. Data (mean ± SEM) are from 4 to 5 mice per group and are representative of two independent experiments. * indicates p=0.0079 by the Mann-Whitney test. (F) Flow cytometry analysis of effector or chronically stimulated (chronic) CD4+ T cells labeled with 2-NBDG ex vivo (higher panel) or following anti-CD3/CD-28 antibody stimulation (lower panel). Means ± SEM of MFI from five animals per group and are representative of three independent experiments. (G) Mitostress assay on chronic CD4+ T cells isolated 7 or 21 days isolated from non-irradiated B6 male (H-2b) *Rag2-/- Il2rg-/-* recipients 21 days after reconstitution with purified Marylin CD4+ TCRBV6+ T cells. Data are representative of two independent experiments. (H) 2-DG-mediated inhibition of IFNγ production by anti-CD3/CD28-stimulated chronic CD4+ T cells. Data (mean ±SEM) were calculated from five replicated cultures and are representative of two independent experiments. (I) Ex vivo expression of inhibitory receptors 2B4, LAG3 and PD-1 by naive Marilyn CD4+ T cells or chronic CD4+ T cells purified from non-irradiated B6 male (H-2b) Rag2-/- Il2rg-/- recipients 7 or 21 days after reconstitution with purified Marylin CD4+ TCRBV6+ T cells. Data are representative of two independent experiments.

DOI: https://doi.org/10.7554/eLife.30938.004

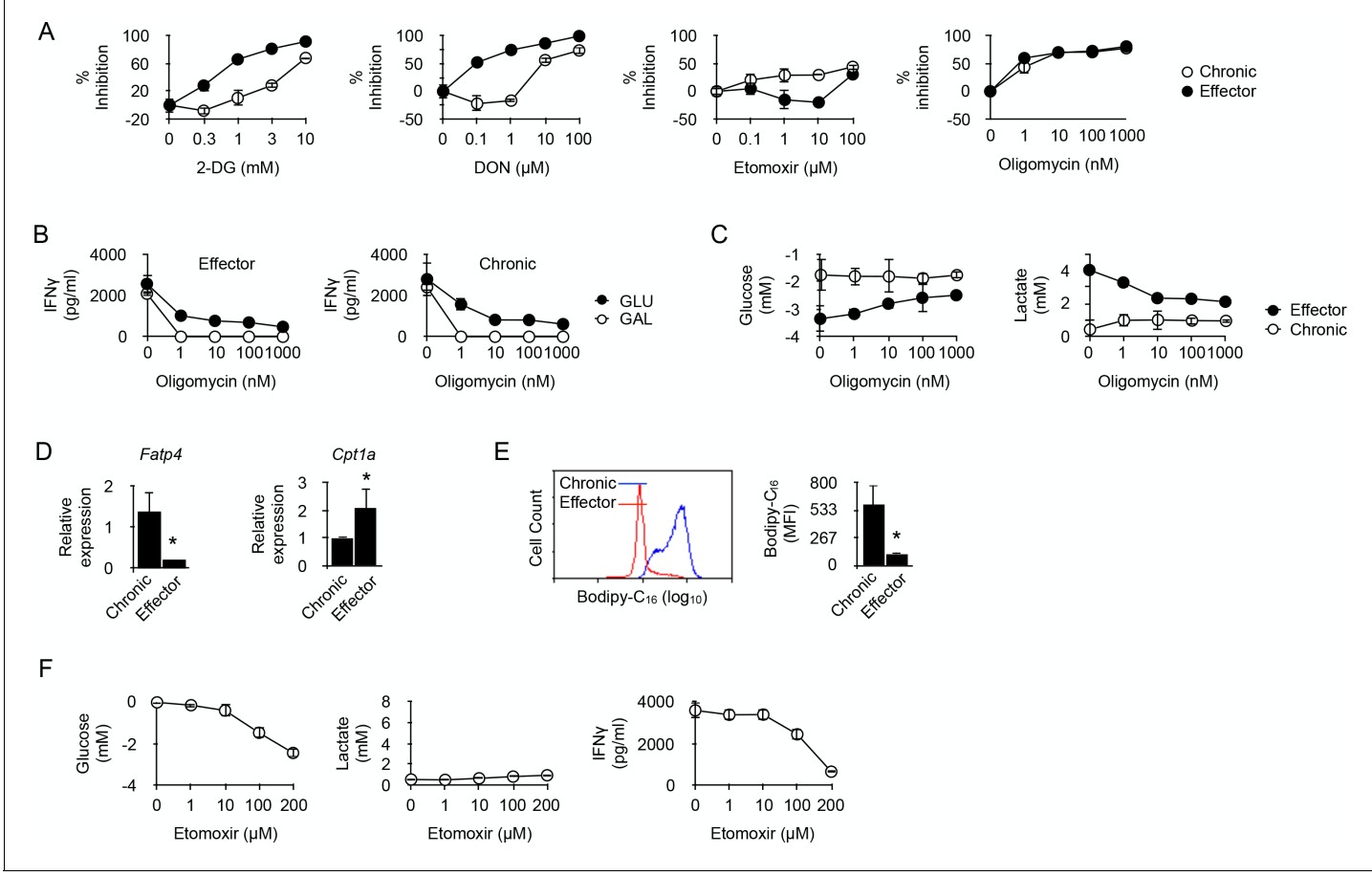

**Figure 4.** Fatty acid oxidation supports the energy demand required for IFNγ production in long-term chronic CD4[+] T cells. (A) Effector or chronic CD4[+] T cells were stimulated with anti-CD3/CD28 antibodies with or without 2-DG, DON, Etomoxir or oligomycin. After 48 hr, culture supernatants were harvested and IFNγ concentrations were determined by ELISA. Percent inhibition were calculated as described in Materials and methods. Data (mean ± SEM) were calculated from five replicated cultures and are representative of three independent experiments. (B) IFNγ production by effector (left) and chronic (right) T cells stimulated with anti-CD3/CD28 antibodies for 48 hr in medium containing glucose or galactose, in the presence of oligomycin. Data (mean ± SEM) were calculated from five replicated cultures and are representative of three independent experiments. (C) Glucose consumption (left) and lactate production (right) in cultures of effector or chronic T cells stimulated with anti-CD3/CD28 antibodies for 48 hr in the presence of oligomycin. Data (mean ± SEM) were calculated from five replicated cultures and are representative of three independent experiments. (D) Ex vivo relative expression of FATP4 and CPT1a mRNA in chronic or effector T cells. Data (mean ± SEM) were calculated from five mice per group and are representative of two independent experiments. * indicates p=0.0079 by the Mann-Whitney test. (E) Bodipy-C16 uptake by chronic or effector T cells. Data (mean ± SEM) were calculated from five replicated cultures and are representative of two independent experiments. * indicates p=0.0079 by the Mann-Whitney test. (F) Glucose consumption (left panel), lactate (middle panel) and IFNγ production (right panel) by chronic CD4[+] T cells stimulated for 48 hr by anti-CD3/CD28 antibodies in the presence of FAO inhibitor Etomoxir. Data were calculated from five replicated cultures and are representative of two independent experiments.

DOI: https://doi.org/10.7554/eLife.30938.006

observations led to the conclusion that H-Y-specific CD4[+] T cells submitted to chronic antigen exposure became increasingly dysfunctional with time, expressing higher levels of inhibitory receptors and failing to engage glycolysis to support effector function.

In cancer cells, glutamine can compensate for glucose depletion in terms of energy generation and synthesis of precursors for anabolic pathways (*Dang, 2010*). In T cells, activation is coupled with rapid glutamine uptake, and failure to transport glutamine throughout the plasma membrane impairs the induction of Th1 and Th17 cells and attenuates inflammatory T cell responses (*Nakaya et al., 2014*). Thus, in glycolysis-deficient chronic CD4[+] T cells, glutaminolysis could represent an alternative pathway on which to base the metabolic requirements for IFNγ synthesis. 6-diazo-5-oxo-L-norleucine (DON), which antagonizes glutamine and is used as a specific inhibitor of

glutamine-utilizing enzymes (*Ahluwalia et al., 1990*), was a very efficient inhibitor of IFNγ synthesis in effector T cells (*Figure 4A*), confirming the essential role played by glutaminolysis in T cell function (*Nakaya et al., 2014*; *Wang et al., 2011*). DON was, however, much less efficient at limiting IFNγ production by chronic T cells (*Figure 4A*). Thus, chronic CD4$^+$ T cells rely much less on glutamine catabolism than effector T cells to carry out their function.

ATP is required for protein synthesis. Therefore, the possible involvement of oxidative metabolism, the main source of cellular ATP, in restoring the function of chronic CD4$^+$ T cells was investigated. T cells were stimulated in the presence of oligomycin, a potent inhibitor of mitochondrial ATP synthase (*Jastroch et al., 2010*). As depicted in *Figure 4A*, oligomycin equally inhibited IFNγ production in effector or chronic CD4$^+$ T cells, indicating that IFNγ synthesis in chronic CD4$^+$ T cells was supported by OXPHOS. This was further investigated by replacing glucose by galactose in the culture medium. Galactose is known to force cells to rely exclusively on OXPHOS for ATP synthesis (*Chang et al., 2013*; *Le Goffe et al., 1999*). As shown in *Figure 4B*, oligomycin fully abolished IFNγ secretion by effector or chronic CD4$^+$ T cells in glucose-free, galactose-containing medium, demonstrating that both cell types can use OXPHOS alone to support IFNγ synthesis. It is known that, when unable to operate OXPHOS, T cells use glycolysis to produce ATP for effector function (*Chang et al., 2013*). This was indeed observed for effector CD4$^+$ T cells that produced lactate and consumed glucose after stimulation in glucose-rich medium in the presence of oligomycin (*Figure 4C*). Conversely, lactate production and glucose consumption were not detected after stimulation of chronic CD4$^+$ T cells in the presence of oligomycin (*Figure 4C*). Taken together, these last results further supported the conclusion that chronic T cells are unable to engage glycolysis for the production of IFNγ.

A recent report indicated that alloreactive T cells responsible for GVHD in irradiated recipients require fatty acid oxidation (FAO) to support their activation and effector function (*Byersdorfer et al., 2013*). We thus investigated whether chronic CD4$^+$ T cells also relied on FAO for IFNγ production. Etomoxir, an irreversible O-carnitine palmitoyltransferase-1 (CPT-1) inhibitor (*Xu et al., 2003*), had a significant inhibitory effect on IFNγ production by chronic T cells that was not observed in effector T cells (*Figure 4A*). Thus, effector function stimulated by anti-CD3-mediated activation would at least partly depend on FAO in chronic CD4$^+$ T cells. Cells that use FAO for energy supply must have the capacity to transport fatty acids to the mitochondria. The long-chain fatty acid transport protein 4 (FATP4) exhibits acyl-CoA synthetase activity and has been proposed, in addition to its function as a transporter, to facilitate uptake of external fatty acids indirectly by mediating their esterification to CoA (*Hall et al., 2005*). FATP4 was identified in a previous RNA microarray analysis among the strongest genes expressed in anti-male chronic CD4$^+$ T cells (three fold; p<0,0004) (*Zhang et al., 2015*). As depicted in *Figure 4D*, FATP4 strong expression was confirmed in chronic T cells when compared to expression in effector T cells, suggesting that import of free fatty acids from their external environment was an important process for chronic T cells. This hypothesis was confirmed by the observation that chronic T cells had the capacity to acquire substantially more fatty acids, such as palmitate, than effector T cells (*Figure 4E*). We also analyzed the expression of CPT1A, responsible for transferring long chain fatty acid acyl-CoA from cytosol into the intermembrane space of mitochondria (an essential step in FAO). Though expressed by both T cell types, CPT1A was significantly higher in effector T cells (*Figure 4D*). Taken together these observations supported the conclusion that FAO was an important metabolic pathway used by chronic T cells for producing IFNγ. This was also supported by the previous observation that the gene coding for acyl-CoA synthetase, bubblegum family, member 1 (ACSBG1), a protein known for its acyl-CoA synthetase activity in the oxidation of long-chain fatty acid, was one of the highest differentially expressed genes in chronic T cells (11.6 fold; p<0.0001) compared to naive T cells (*Zhang et al., 2015*). FAO is known to impair the glycolytic pathway mostly through inhibition of pyruvate dehydrogenase (PDH) and/or glucose uptake and 6-phosphofructo-1-kinase (PFK-1) (*Hue and Taegtmeyer, 2009*; *Bowker-Kinley et al., 1998*). Therefore, we evaluated the effect of FAO inhibition by Etomoxir on glucose metabolism in chronic T cells. As expected, adding Etomoxir in cultures of anti-CD3/CD28-stimulated chronic T cells increased substantially their consumption of glucose (*Figure 4F*). Importantly, however, increased glucose uptake did not restore their capacity to produce IFNγ nor was this accompanied by lactate production, demonstrating once more that cytokine synthesis in chronic T cells was carried out independently from glucose metabolism.

## Genetic absence of PD-1 does not promote a functional metabolic switch towards glycolysis in chronic CD4[+] T cells

Since PD-1 function has been recently linked to metabolism regulation in T cells (*Patsoukis et al., 2015*; *Gubin et al., 2014*), whether PD-1 engagement was required for modulating the functional metabolic profile of chronic CD4[+] T cells was investigated. IFNγ was detected in the supernatant of cultures containing whole spleen cells isolated from male recipients that received anti-male *Pdcd1*[-/-] CD4[+] T cells (*Figure 5A*). IFNγ production was abrogated by adding antibodies blocking the interaction between anti-male TcR and MHC class II/male peptide complexes present at the surface of male splenic APCs, demonstrating that IFNγ was indeed produced following male antigen recognition by T cells (*Figure 5A*). Thus, as expected, the genetic absence of PD-1 prevented functional regulation in anti-male-specific CD4[+] T cells in lymphopenic recipients and promoted IFNγ secretion. Both chronic *Pdcd1*[+/+] and *Pdcd1*[-/-] T cells had lower basal ECAR, consistent with a missing capacity to mobilize glycolysis for IFNγ production (*Figure 5B*). On the contrary, basal OCR was significantly more intense in chronic *Pdcd1*[-/-] T cells, suggesting that IFNγ produced in the absence of PD-1 regulation could be supported by OXPHOS (*Figure 5C*). Moreover, the low maximal/basal OCR ratio exhibited by *Pdcd1*[-/-] T cells suggested that a great part of OXPHOS was committed to cytokine secretion (*Figure 5C*). The incapacity of chronic *Pdcd1*[-/-] T cells to sustain a high rate of glycolysis was confirmed by the observation that, after activation, cytokine production was not accompanied by the production of lactate (*Figure 5D*). Accordingly, adding 2-DG did only marginally inhibit IFNγ production in chronic *Pdcd1*[-/-] CD4[+] T cells after anti-CD3/CD28 stimulation (*Figure 5E*). Taken together, these observations demonstrated that chronic antigen stimulation in the absence of PD-1-mediated regulation did not program T cell metabolism towards glycolysis to support effector function.

We also investigated whether, like their *Pdcd1*[+/+] counterparts, chronic *Pdcd1*[-/-] CD4[+] T cells used FAO to support IFNγ production. As indicated in *Figure 5E*, PD-1-deficient chronic T cells were as sensitive as PD1-sufficient T cells to inhibition by Etomoxir or oligomycin. Therefore, fatty acids appear an essential source of energy for the synthesis of IFNγ in chronic male-specific CD4[+] T cells, regardless of their expression of PD1. Ex vivo analysis of FATP4 expression suggested that both *Pdcd1*[+/+] and *Pdcd1*[-/-] CD4[+] T cells appeared to have a similar capacity to acquire long chain fatty acids from their environment (*Figure 5F*). Accordingly, both types of T cells were able to incorporate equivalent amount of Bodipy (*Figure 5G*). However, *Pdcd1*[-/-] CD4[+] T cells displayed high CPT1A expression (*Figure 5F*). Taken together, these results support the notion that, though displaying the same capacity to import fatty acid from their direct environment, *Pdcd1*[+/+] and *Pdcd1*[-/-] CD4[+] T cells would differ in the way they import fatty acids in the intermembrane mitochondrial space where FAO takes place. Thus, fatty acid entry into mitochondria could represent the limiting factor for IFNγ production in chronic CD4[+] T cells.

## PD-1 protects against reactive oxygen species (ROS)-mediated cell death and impaired functional fitness in chronic CD4[+] T cells

Our proteomic comparison before and after chronic exposure to antigen revealed a significant deficit in the expression of proteins important for oxidant detoxification in chronic T cells (*Table 1*) (*Hanschmann et al., 2013*). These results suggested that chronic antigen stimulation might render T cells more sensitive to oxidative stress. Oxidative stress is mediated, in part, by reactive oxygen species (ROS) produced by multiple cellular processes and controlled by cellular antioxidant mechanisms (*Hanschmann et al., 2013*). Mitochondrial production of ROS was detected ex vivo in chronic CD4[+] T cells. Though ROS production by *Pdcd1*[+/+] T cells was modest, it was very significant in chronic *Pdcd1*[-/-] T cells (*Figure 6A*). As expected, higher ROS production in chronic *Pdcd1*[-/-] T cells was correlated with higher mitochondrial depolarization (*Figure 6B*). Thus, ROS production appeared to be controlled by PD-1 in chronic T cells. It is well established that, upon antigenic stimulation, T cells produce mitochondrial ROS that regulate secondary signaling molecules to bring about optimal TcR activation (*Devadas et al., 2002*; *Kaminski et al., 2010*). However, excessive mitochondrial ROS production can also lead to T cell death, and antioxidant chemicals can improve the survival of activated T cells (*Dumont et al., 1999*; *Cossarizza et al., 1997*). As shown in *Figure 6C*, N-acetyl-cysteine (NAC) significantly increased the viability of stimulated chronic T cells. Consistent with their higher levels of mitochondrial ROS, chronic *Pdcd1*[-/-] T cells were less sensitive

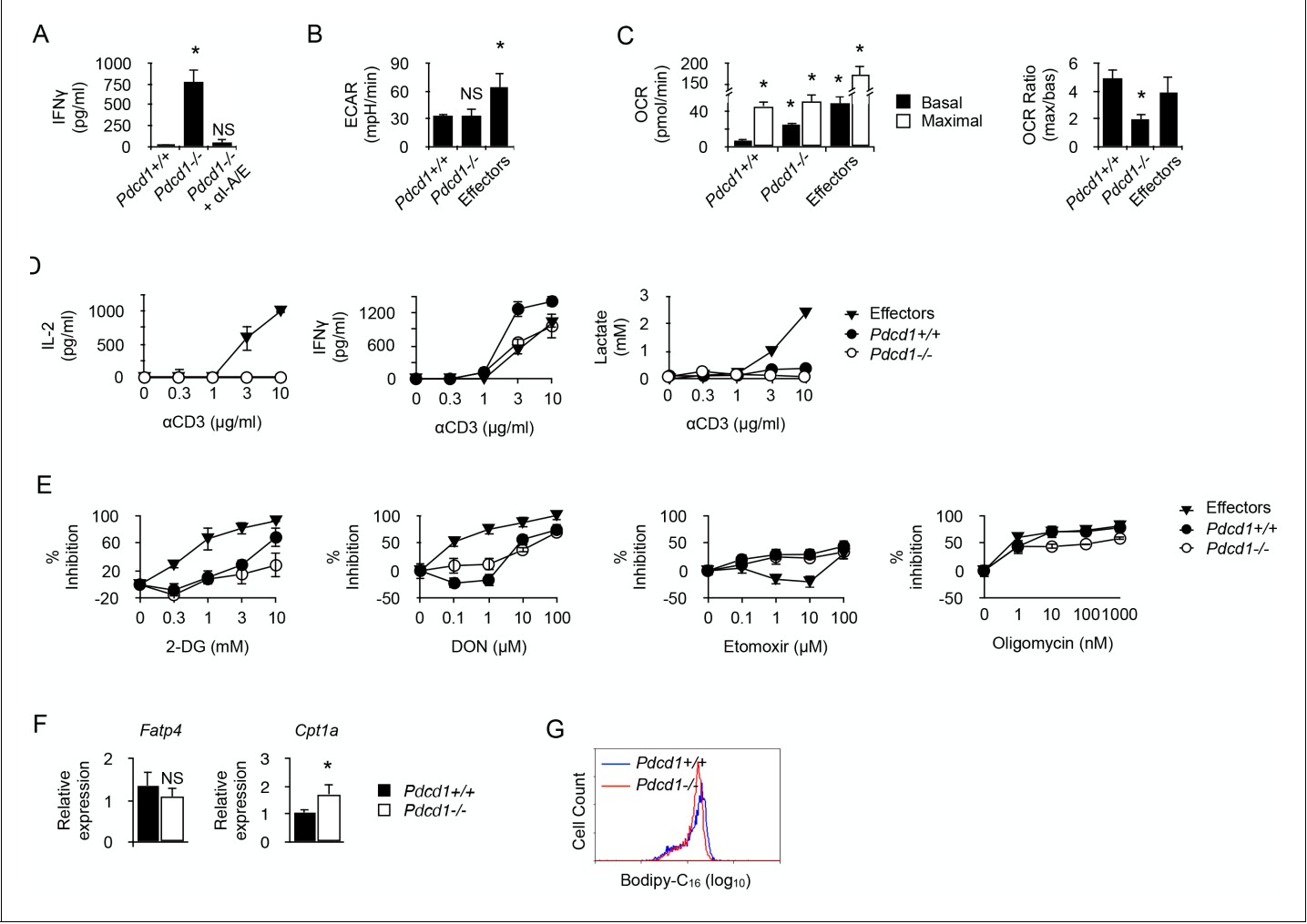

**Figure 5.** PD-1 is not required for the induction and maintenance of the metabolic profile developed by chronic CD4[+] T cells. (**A**) IFNγ production in cultures of whole spleen cells isolated from male *Rag2[-/-] Il2rg[-/-]* B6 recipients adoptively transferred with anti-male *Pdcd1[+/+]* or *Pdcd1[-/-] Rag2[-/-]* Marilyn T cells for 21 days. Anti-I-A/I-E antibodies were added as indicated. Data (mean ± SEM of five replicated cultures) are representative of three independent experiments. * indicates p=0.0079 (compared to *Pdcd1[+/+]*) by the Mann-Whitney test. NS indicates non-significant. (**B**) Basal ECAR of chronic *Pdcd1[+/+]* or *Pdcd1[-/-]* CD4[+] T cells and effector CD4[+] T cells. Data (mean ± SEM of five replicated cultures) are representative of two independent experiments. * indicates p=0.0079 and NS indicates non-significant by the Mann-Whitney test. (**C**) Basal and maximal OCR of chronic *Pdcd1[+/+]* or *Pdcd1[-/-]* CD4[+] T cells and effector CD4[+] T cells. (left panel). Maximal/basal OCR ratios (right panel). Data (mean ± SEM of five replicated cultures) are representative of two independent experiments. * indicates p=0.0079 by the Mann-Whitney test. (**D**) Chronic *Pdcd1[+/+]* or *Pdcd1[-/-]* CD4 +T cells or effector CD4[+] T cells were stimulated with anti-CD3/CD28 antibodies. After 48 hr, culture supernatants were harvested and lactate, IL-2 and IFNγ concentrations were determined as described in Material and methods. Data (mean ± SEM from five replicated cultures) are representative of three independent experiments. (**E**) Chronic *Pdcd1[+/+]* or *Pdcd1[-/-]* CD4[+] T cells or effector CD4[+] T cells were stimulated with anti-CD3/CD28 antibodies in the presence of 2-DG, DON, Etomoxir or oligomycin. After 48 hr, culture supernatants were harvested and IFNγ concentrations were determined by ELISA. Percent inhibition was calculated as described in Materials and methods. Data (mean ± SEM) were calculated from five replicated cultures and are representative of three independent experiments. (**F**) Ex vivo relative expression of FATP4 and CPT1a mRNA in *Pdcd1[+/+]* or *Pdcd1[-/-]* chronic T cells. Data (mean ± SEM) were calculated from five replicated cultures and are representative of three independent experiments. * indicates p=0.0079. (**G**) Bodipy-C16 uptake by in *Pdcd1[+/+]* or *Pdcd1[-/-]* chronic T cells. Data are representative of two independent experiments.

DOI: https://doi.org/10.7554/eLife.30938.007

to NAC-mediated inhibition of cell death (*Figure 6C*). Thus, chronic antigen stimulation in the absence of PD-1-mediated inhibition promoted enhanced production and accumulation of ROS in mitochondria and rendered T cells more prone to cell death. It should be noted that this last conclusion is in agreement with several observations that deletion/blockade of PD-1 is associated with decreased survival of virus-specific CD8[+] T cells during chronic viral infection (*Odorizzi et al., 2015*;

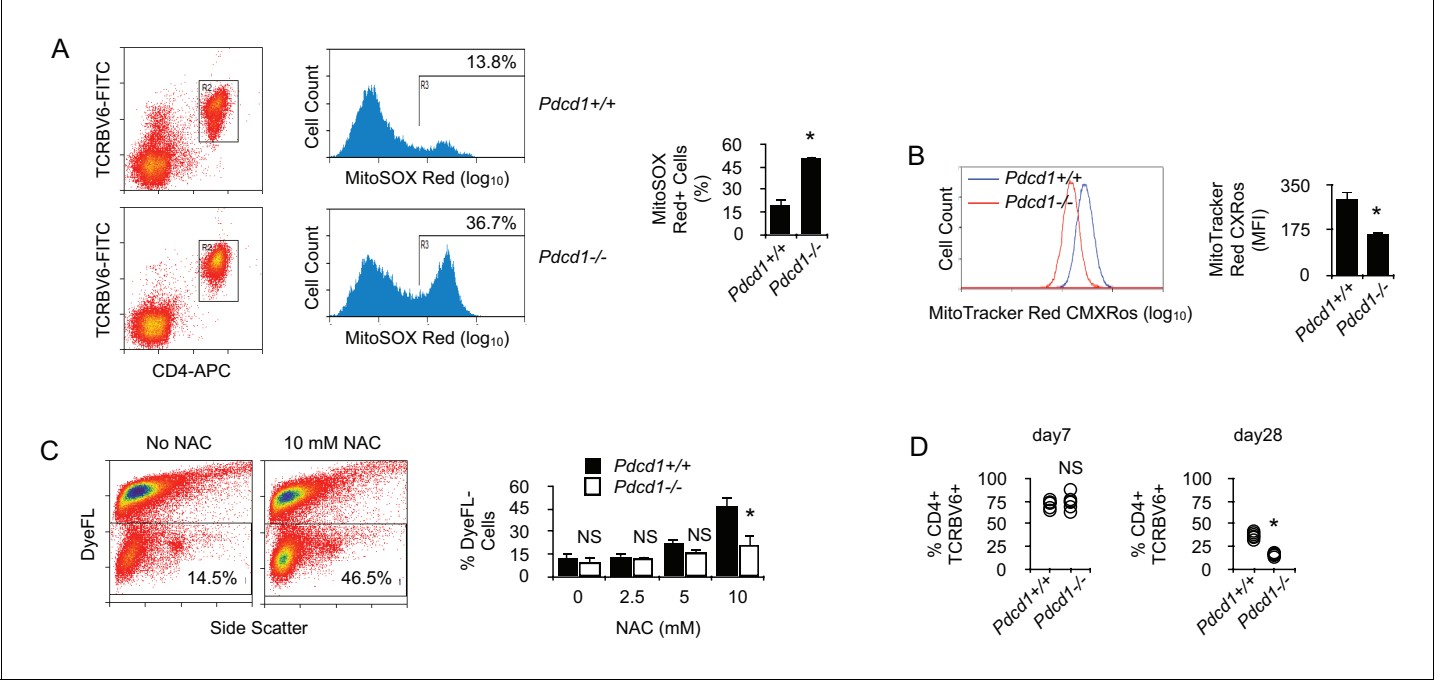

**Figure 6.** By limiting T cell metabolism, PD-1 prevents activation-dependent mitochondrial ROS production and cell death. (**A**) Flow cytometry analysis of chronic *Pdcd1+/+* or *Pdcd1-/-* CD4+ T cells for intra-mitochondrial ROS production. Ex vivo spleen cells were labeled with MitoSOX Red and analyzed by flow cytometry. Gating was carried out on TCRBV6+ CD4+ cells as indicated. Mean ± SEM of percentage of positive cells from TCRBV6+ CD4+ T cells isolated from 5 mice is shown. Data are representative of two independent experiments. * indicates p=0.0079. (**B**) Flow cytometry analysis of chronic *Pdcd1+/+* or *Pdcd1-/-* CD4+ T cells for their mitochondrial membrane potential. Ex vivo spleen cells were labeled with MitoTracker Red CMXRos and analyzed by flow cytometry. Analysis was gated on TCRBV6+ CD4+ cells as in (**A**). Mean ± SEM of MFI from TCRBV6+ CD4+ T cells isolated from 5 mice per group is shown. Data are representative of two independent experiments. * indicates p=0.0079 by the Mann-Whitney test. (**C**) Cell viability of purified chronic *Pdcd1+/+* or *Pdcd1-/-* CD4+ T cells after anti-CD3/CD28 stimulation for 24 hr culture with or without NAC. Mean ± SEM of percentage of viable cells (DyeFL-negative) in TCRBV6+ CD4+ T cells isolated from 5 mice per group is shown. Data are representative of two independent experiments. * indicates p=0.0079 by the Mann-Whitney test. NS indicates non-significant. (**D**) Frequencies (%) of anti-male CD4+ TCRBV6+ T cells in the spleen of lymphopenic male recipients. Cells were analyzed 7 and 28 days after adoptive transfer. Mean ± SEM of percentage of TCRBV6+ CD4+ T cells isolated from 6 mice per group is shown. Data are representative of two independent experiments. * indicates p=0.0079 by the Mann-Whitney test. NS indicates non-significant.

DOI: https://doi.org/10.7554/eLife.30938.008

*Staron et al., 2014*). Accordingly, we also observed that, after transfer into lymphopenic male recipients, splenic male-specific *Pdcd1-/-* CD4+ T cells, though exhibiting an early expansion identical to that of *Pdcd1+/+* CD4+ T cells, were less present at a later time point (*Figure 6D*). Remarkably, ROS inhibition by NAC also improved the capacity of chronic T cells to produce IFNγ upon stimulation (*Figure 7A and B*). Adding NAC to PMA/ionomycin-stimulated T cells increased indeed the frequency of IFNγ-secreting cells (*Figure 7A*), as well as the amount of IFNγ produced per cell (*Figure 7B*). Again, consistent with their higher production of mitochondrial ROS, NAC did not augment IFNγ production in *Pdcd1-/-* CD4+ T cells as much as in *Pdcd1+/+* CD4+ T cells (*Figure 7B*). Thus, taken collectively, these data support the conclusion that one of the main roles played by PD-1 in chronic CD4+ T cells is to limit the production of mitochondrial ROS, thereby promoting cell viability and functional fitness.

Last, we wished to identify how ROS could modulate IFNγ production in chronic T cells. One mechanistic explanation would be that elevated ROS could inactivate mTORC1 (*Li et al., 2010*), leading to reduced protein translation. In order to investigate this possibility, we tested mTORC1 signaling by Western blots in ex vivo isolated chronic CD4+ T cells. As depicted in *Figure 7C*, phosphorylation of mTORC1 targets, translation initiation factor 4E-binding protein 1 (4EBP1) and ribosomal protein S6, was more important in high ROS producer T cells, namely *Pdcd1-/-* CD4+ T cells. Moreover, anti-CD3/CD28 stimulation induced similar levels of phosphorylation of both 4EBP1 and

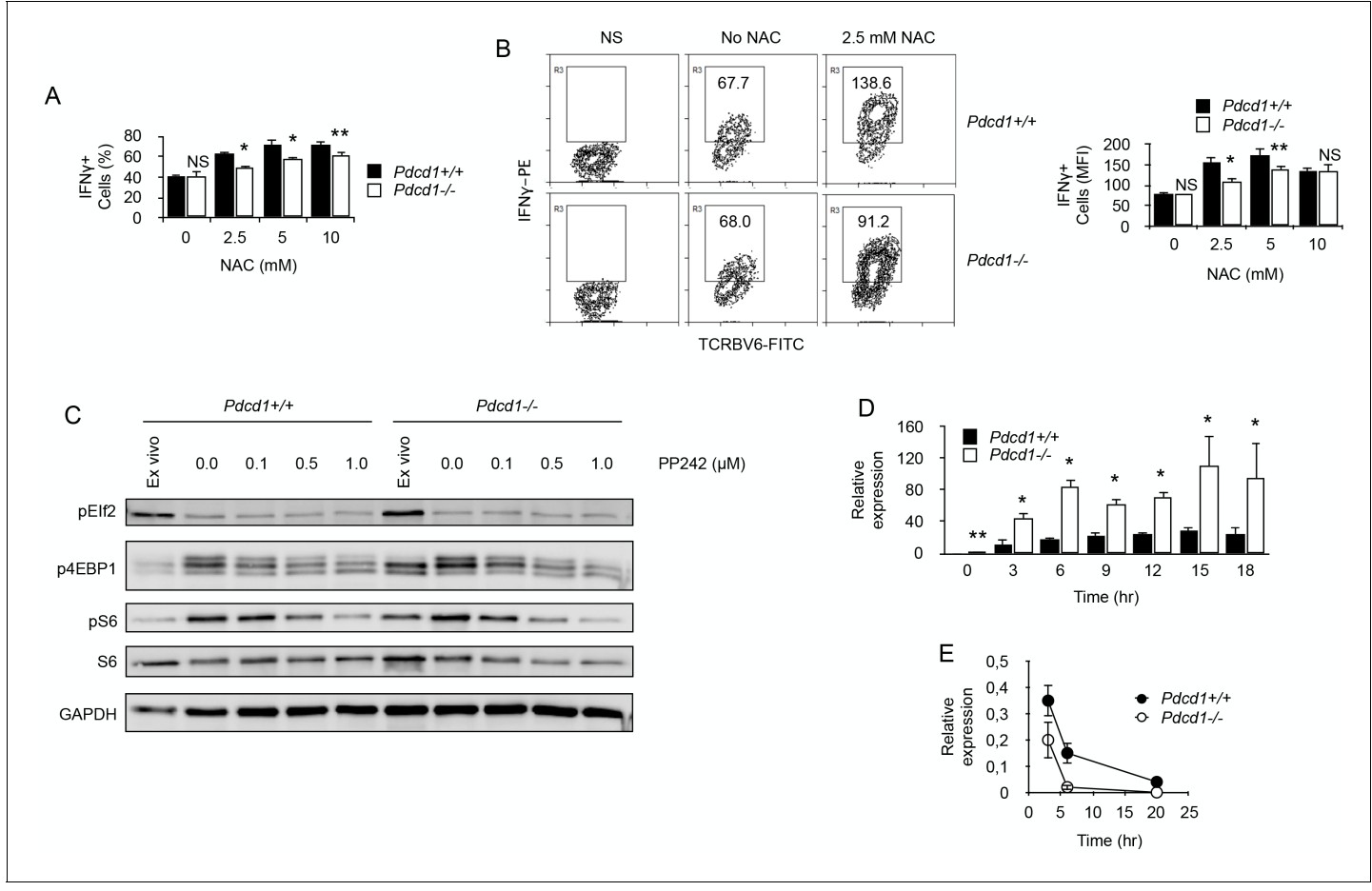

**Figure 7.** By limiting ROS production, PD-1 maintains functional fitness in chronically stimulated CD4+ T cells. (**A**) Frequencies (%) of IFNγ-producing cells among purified chronic Pdcd1+/+ or Pdcd1−/− CD4+ T cells stimulated with anti-CD3/CD28 antibodies for 24 hr with or without NAC. After adding Brefeldin for the last 4 hr, cells were labelled for intracellular IFNγ. Mean ± SEM from T cells isolated from 5 mice per group is shown. Data are representative of two independent experiments. * and ** indicate p=0.0079 and p=0.0286, respectively, by the Mann-Whitney test. NS indicates non-significant. (**B**) Relative IFNγ production among purified chronic Pdcd1+/+ or Pdcd1−/− CD4+ T cells stimulated with anti-CD3/CD28 antibodies for 24 hr with or without NAC as in D. After adding Brefeldin for the last 4 hr, cells were labelled for intracellular IFNγ. Mean of MFI ± SEM from T cells isolated from 5 mice per group is shown. Data are representative of two independent experiments. * and ** indicate p=0.0079 and p=0.0286, respectively, by the Mann-Whitney test. NS indicates non-significant. (**C**) Western blotting analysis of mTORC1-dependent control of protein translation. Phosphorylation status of mTORC1 targets S6 protein and 4EBP1 was analysed in chronic Pdcd1+/+ or Pdcd1−/− CD4+ T cells ex vivo or after stimulation with anti-CD3/CD28 antibodies for 48 hr with or without mTOR inhibitor PP242. Analysis of Elf2 phosphorylation was used as a specificity control. Data are representative of three independent experiments. (**D**) Relative expression of IFNγ mRNA in chronic Pdcd1+/+ or Pdcd1−/− CD4+ T cells stimulated by anti-CD3/CD28 antibodies for the indicated periods of time. Data (mean ± SD) are representative of two independent experiments including 4 mice per group. * and ** indicate p=0.0079 and p=0.0286, respectively. (**E**) Analysis of IFNγ mRNA decay in chronic Pdcd1+/+ or Pdcd1−/− CD4+ T cells stimulated by anti-CD3/CD28 antibodies. Data (mean ± SD) are one experiment including 4 mice per group.

DOI: https://doi.org/10.7554/eLife.30938.009

S6 in Pdcd1+/+ or Pdcd1−/− CD4+ T cells and mTOR inhibition by PP242 reduced to the same extent 4EBP1 and S6 phosphorylation in both T cell populations (**Figure 7C**). These results demonstrate that mTORC1 signaling worked equally well in Pdcd1+/+ or Pdcd1−/− chronic CD4+ T cells and are not consistent with the hypothesis that IFNγ production in chronic T cells is limited by ROS inhibitory activity on mTOR signaling and protein synthesis. Another possibility is that IFNγ production would be regulated by ROS at RNA level. Surprisingly, after stimulation with anti-CD3/CD28 antibodies, IFNγ mRNA was consistently more abundant in high ROS producer Pdcd1−/− chronic CD4+ T cells (**Figure 7D**). This was not expected since, most of the time after stimulation, IFNγ production was reduced among Pdcd1−/− chronic CD4+ T cells compared to Pdcd1+/+ chronic CD4+ T cells (**Figures 5D, 7A and B**). RNA damage has been reported to increase under oxidative stress and

ROS are a major source of damage to cellular components including RNA molecules (*Fiala et al., 1989*; *Shen et al., 2000*; *Hofer et al., 2005*). Since ROS production was more important in *Pdcd1*[-/-] chronic CD4[+] T cells, we postulated that RNA oxidation by ROS should be more intense in these cells than in *Pdcd1*[+/+] chronic CD4[+] T cells. High levels of RNA oxidation are normally not tolerated by cells, and oxidized RNA is actively removed from total RNA by specific mechanisms (*Li et al., 2006*). Therefore, we measured IFNγ mRNA half/life in chronic T cells by blocking cellular transcription with actinomycin D. As seen in *Figure 7E*, RNA decay was more pronounced in *Pdcd1*[-/-] chronic CD4[+] T cells. This observation is consistent with the idea that increased RNA oxidation limited IFNγ production in these cells.

## Discussion

In the present study, we identified the metabolic profile that characterizes CD4[+] T cells submitted to persisting antigen signaling. Overall, chronic T cells exhibited a general deficit in metabolism characterized by a reduced OXPHOS and the incapacity to engage glycolysis. The observation that effector function does not rely on glycolysis in chronic CD4[+] T cells is in sharp contrast with the highly glycolytic metabolic profile generally observed in effector T cells which fail to produce IFNγ in the absence of glycolysis (*Chang et al., 2013*; *Gubser et al., 2013*; *Cham et al., 2008*; *Cham and Gajewski, 2005*). Moreover, recent reports clearly demonstrated glycolysis as the predominant metabolic pathway used by donor T cells to promote GVHD in allogeneic HSC transplanted recipients (*Saha et al., 2013*; *Nguyen et al., 2016*). These diverging results might be explained by the different time laps during which T cells were submitted to chronic stimulation. Indeed, the present study and others (*Bengsch et al., 2016*; *Saha et al., 2013*; *Nguyen et al., 2016*) identified the function of T cells responsible in early chronic stimulation (day 6–10 after adoptive transfer) as highly sensitive to glycolysis inhibition. On the contrary, late chronic CD4[+] T cells (day 21 after adoptive transfer), that were characterized in our study, failed to up-regulate glycolysis and relied mostly on OXPHOS to support their function after activation. Interestingly, several recent studies have reported a similar metabolic defect in CD4[+] T cells isolated from patients affected by chronic autoimmune diseases (*Yang et al., 2013*; *Wahl et al., 2010*). Thus, failing to engage glycolysis for effector function could represent a general feature of long-term chronically-stimulated CD4 T cells.

Our results contrast with those obtained by others in the exhaustion of LCMV-specific CD8[+] T cells where PD-1 altered both glycolytic and mitochondrial bioenergetics (*Bengsch et al., 2016*). In that model, reviving T cell function by neutralization of PD-L1 resulted in reduced mitochondrial mass, increased glucose uptake and glycolysis in virus-specific T cells (*Bengsch et al., 2016*). This is sharp contrast with our results where blocking PD-1 restored function but not glycolysis and was supported by increased OXPHOS activity. One of the major differences between T cells in our system and in the chronic LCMV infection model is that, in LCMV-specific exhausted CD8[+] T cells, PD-1 restores both function and proliferation (*Barber et al., 2006*). We have previously shown that PD-1 blockade in our model does not restore proliferation but revives partly function (IFNγ production, not IL-2). Thus, male-specific chronic CD4[+] T cells in our study appear to have reach a deeper state of unresponsiveness, as indicated by higher expression of PD-1 and other inhibitory receptors, where glycolysis would not be anymore available to sustain T cell function after PD-1 blockade. This conclusion is also supported by our observation that PD-1-mediated control on glycolytic metabolism is also present in the early stage of exhaustion in male-specific chronic CD4[+] T cells. Finally, as discussed below, chronic antigen stimulation in our adoptive transfer experiments is presumably achieved without inflammatory signals, contrary to what is seen in chronic viral infection. Thus, chronic antigen recognition in the absence of costimulatory signals might induce a different metabolic program involving the complete suppression of T cell glycolytic capacity. Interestingly, rendering T cells anergic through TcR stimulation alone not only prevents upregulation of glycolytic metabolism, but also prevents it from being upregulated in future stimulations, even those delivered with costimulation, suggesting that these cells are metabolically anergic (*Zheng et al., 2009*).

Surprisingly, PD-1-mediated inhibition of TcR-signaling did not appear to be required for the induction and maintenance of the non-glycolytic profile displayed by late chronic male-specific CD4[+] T cells. This was not expected since PD-1 engagement by its ligand PD-L1 has been shown to block in vitro effector cell differentiation in human T cells by inhibiting glycolysis and promoting FAO (*Patsoukis et al., 2015*). Moreover, recent data indicated that PD-1 was critical for the regulation of

glycolytic metabolism in T cells responsible for acute GVHD (*Saha et al., 2013*). It should be pointed out that chronic T cells in our model express multiple inhibitory receptors at their surface, thereby providing potential compensatory regulatory mechanisms for metabolic control. This hypothesis is supported by the observation that engagement of the inhibitory receptor CTLA-4 at the surface of human T cells has also been shown to inhibit glycolysis (*Patsoukis et al., 2015*). Because CTLA-4 is expressed at the surface of chronic T cells in our model (*Noval Rivas et al., 2009*), it could be responsible for maintaining low levels of glycolysis in the absence of PD-1. At this stage, it is interesting to note that Schietinger and coworkers have shown tumor-specific CD8$^+$ T cell exhaustion to be solely driven by antigenic stimulation, mostly independently from the activity of inhibitory receptors like PD-1 (52). To explain their observation, the authors put forward the fact that, contrary to virus-specific T cells, unresponsiveness in tumor-specific T cells is initiated and maintained in the absence of inflammation. In that context, T cells see their antigen presented by non-activated APC that do not express the co-stimulatory molecules CD80 and CD86 and cannot deliver signals, via CD28 engagement, required for full T cell activation (*Schietinger et al., 2016*). This type of unresponsiveness might develop independently of the activity of inhibitory receptors such as PD-1 (52). Like tumor-specific T cells, male-specific CD4$^+$ T cells in our system encounter antigen in a non-inflammatory context. Our results would indicate that T cell dysfunction induced by chronic antigen encounter in a non-inflammatory environment would involve the establishment of a specific non-glycolytic metabolic program. Interestingly, it was very recently observed that sustained glycolytic function in T cells required CD28 signaling (*Menk et al., 2018*).

It has been previously shown that PD-1 repressed the expression of PGC-1a, a key transcriptional regulator of genes controlling energy metabolism and mitochondrial biogenesis, and enhancing PGC-1a expression reversed mitochondria-associated metabolic perturbations in early stage of T cell exhaustion (*Bengsch et al., 2016*). In a previous analysis of gene expression by microarray, PGC-1a was not found differently expressed by chronic CD4$^+$ T cells or naive T cells (*Zhang et al., 2015*). However, PPARγ, with which PGC-1a interacts to coordinate the expression of genes involved in mitochondrial function (*Schreiber et al., 2004*), was down-regulated (five fold; p<0;0001) in chronic CD4$^+$ T cells compared to naive T cells (*Zhang et al., 2015*). This is consistent with the exhausted phenotype and the decreased respiratory activity displayed by chronic T cells. However, our observation that *Pdcd1$^{+/+}$* or *Pdcd1$^{-/-}$* chronic T cells did not differ in their maximal respiratory capacity would not support the idea that, in late chronic T cells, PD-1 mediates its inhibitory effect on T cell function through the direct regulation of genes involved in energy metabolism and mitochondrial biogenesis.

The last important and somehow surprising result of our study is the observation that, in chronic CD4$^+$ T cells, PD-1 maintains T cell functional fitness and viability by limiting the intensity of oxidative metabolism and ROS production. This is in sharp contrast with previous observations made by several groups where PD-1-mediated regulation of T cell function was shown to repress mitochondrial metabolism thereby promoting the production of ROS and potentially cell death in some but not all T cells activated by antigen (*Bengsch et al., 2016*; *Tkachev et al., 2015*). However, our data indicate that chronic CD4$^+$ T cells mostly rely on OXPHOS for effector function after stimulation. In the absence of glycolysis, excessive OXPHOS activity could lead to overload the electron transport chain and, consequently, to increase mitochondrial ROS levels that may result in significant damage to cell structures and viability. This conclusion is supported by the observation that, in our study, PD-1 regulatory activity inhibits both mitochondrial activity and ROS production in chronic T cells. Thus, PD-1 would appear to fulfill two different functions in chronic CD4$^+$ T cells depending on the duration of chronic antigenic exposure (*Figure 8*). On the one hand, during early chronic antigen stimulation, PD-1-mediated inhibition of TcR signaling creates bioenergetics deficiencies to control T cell function. By contrast, our data show that, over prolonged antigen stimulation, PD-1 limits T cell oxidative metabolism in order to maintain functional fitness and cell survival. Moreover, we also show that in chronic CD4$^+$ T cells IFNγ production is impaired by fast mRNA decay. ROS are a major source of damage to cellular components (*Li et al., 2006*). RNA damage has been reported to increase during oxidative stress. Under similar conditions, the levels of oxidative damage in RNA are usually higher than those in DNA, which may impair protein synthesis (*Li et al., 2006*). Therefore, accumulation of RNA damage must be prevented and cells have developed specific mechanisms to remove oxidized RNA molecules (*Li et al., 2006*). Our data support the notion that, in chronic CD4$^+$ T cells, PD-1

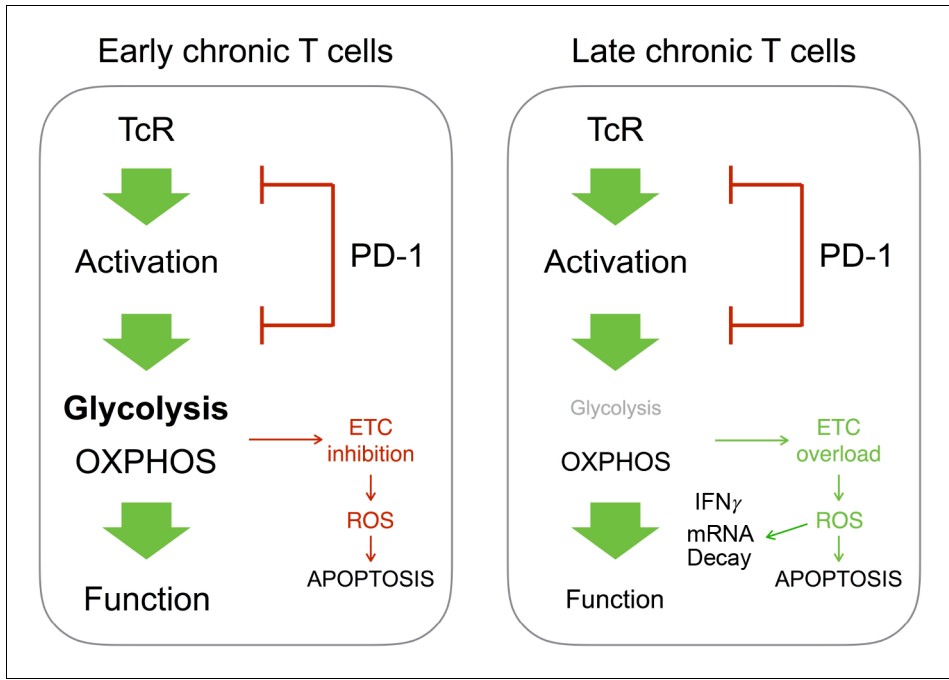

**Figure 8.** Different outcome from PD-1-mediated regulation of T cell metabolism in early or late chronic antigen stimulation. In chronic T cells responsible for early GVHD, PD-1-mediated inhibition of T cell metabolism promotes mitochondrial dysfunction and cell death. In T cells responsible for late GVHD, failure to engage glycolysis after PD-1 blockade stimulates OXPHOS. This leads to overload of the electron transport chain (ETC), ROS production, mRNA decay and cell death.

DOI: https://doi.org/10.7554/eLife.30938.010

increases T cell activation threshold to prevent excessive RNA oxidation and decay, consequently preserving the capacity of T cells to produce inflammatory cytokines such as IFNγ.

Overall, our data are consistent with the concept that duration of antigen encounter modifies the different metabolic programs available to activated CD4[+] T cells for supporting their function. Naive CD4[+] T cells stimulated by antigen undergo a specific differentiation program, mostly relying on glycolysis, that leads to clonal expansion and development of effector function. Sustained antigenic stimulation, however, causes alteration in this differentiation process and forces CD4[+] T cells to enter a program associated with functional unresponsiveness. This program involves the expression of cell surface inhibitory receptors, such as PD-1, whose engagement by their ligands present at the surface of MHC class II-expressing APCs inhibits glycolytic and mitochondrial metabolic pathways required for T cell proliferation and effector function. If chronic antigenic stimulation continues after inflammation has resorbed, or if antigen is presented without co-stimulatory signals delivered by APCs that express high levels of CD80 and CD86, CD4[+] T cells will enter a cell-intrinsic program apparently driven solely by continuous antigen exposure, independently of PD-1 engagement. In this context, it is interesting to note that engagement of cell surface CD28, the receptor responsible for delivering CD80/CD86 co-stimulation to T cells, is required for CD8 T cell rescue involving anti-PD-1 therapy (*Kamphorst et al., 2017*). The dysfunctional state imposed by non-inflammatory continued antigen presentation would involve the complete loss of glycolysis as a metabolic pathway available for sustaining cell proliferation and cytokine synthesis and could not be rescued by checkpoint blockade. Thus, although early metabolic dysfunction is still a characteristic of a plastic differentiation state, it ultimately becomes fixed and non-reversible. Our study could have important implications for the control of T cell activity in chronic immune diseases. Our data support the possibility that promoting PD-1-mediated regulation of T cell metabolism might represent an effective therapeutic tool for controlling the early steps of chronic immune pathologies. Paradoxically, they also enlighten the potential therapeutic limitations that blocking PD-1/PD-L1 interactions might face in attempting to restore the function of T cells submitted to long-term chronic antigen stimulation.

## Materials and methods

### Mice

Normal B6 (H-2$^b$) and BALB/c (H-2$^d$) mice were from Charles River. Female *Rag2*$^{-/-}$ Marilyn mice, transgenic for a TCR (*Tcrav1.1*, *Tcrbv6*) specific for the male H-Y peptide NAGFNSNRANSSRSS presented by I-A$^b$, were obtained from O. Lantz (Curie Institute, Paris, France). *Rag2*$^{-/-}$ *Il2rg*$^{-/-}$ B6 mice were obtained from J. P. Di Santo (Pasteur Institute, Paris, France). *Pdcd1*$^{-/-}$ and GFP$^+$ Marilyn mice were obtained by crossing Marilyn mice with *Pdcd1*$^{-/-}$ B6 (BIO Resource Center, RIKEN) and GFP$^+$ B6 (The Jackson Laboratories), respectively. *Cd3e*$^{-/-}$ B6 mice were from The Jackson Laboratories. All mice used in this study were issued from breeding pairs housed in specific pathogen-free conditions (FELASA) at the Institute for Medical Immunology. Experimental animal protocols were performed in accordance with the Animal Care and Use Committee guidelines of the Université Libre de Bruxelles.

### T cell purification

CD4$^+$ T cells were purified from spleen by magnetic-activated cell sorting (Dynal CD4$^+$ T cell negative isolation kit; Invitrogen). For the two-dimensional gel electrophoresis, spleen cells were incubated with FITC-conjugated anti-mouse TCRVβ6 antibody (RR4-7; BD Pharmingen), then with anti-FITC magnetic beads (Miltenyi Biotec) and sorted by magnetic sorting according to the manufacturer's protocol. T cell purity assessed by flow cytometry was >92%.

### Generation of chronically stimulated T cells and effector T cells

Chronically stimulated allogeneic CD4$^+$ T cells were generated by adoptive transfer of purified BALB/c (H-2$^d$) CD4$^+$ T cells (5 × 10$^6$ cells/recipient) into non-irradiated B6 (H-2$^b$) *Rag2*$^{-/-}$ *Il2rg*$^{-/-}$ recipients. H-Y-specific chronic CD4 +T cells were obtained by injecting i.p. anti-male CD4$^+$ T cells (1 × 10$^6$ cells/recipient) from Marylin mice into non-irradiated male B6 *Rag2*$^{-/-}$ *Il2rg*$^{-/-}$ recipients.

For the generation of H-Y-specific effector CD4$^+$ T cells, female Marilyn mice were immunized i.p. with 1 × 10$^7$ irradiated (20 Grays) spleen cells from male CD3ε-deficient mice or male Marilyn mice. The mice were sacrificed 7 days later and effector T cells were purified as indicated above.

### Cell culture

All cultures were in RPMI 1640 medium (with 2 mM L-glutamine without glucose) supplemented with 10% (vol/vol) heat-inactivated dialyzed FCS (Life Technologies) and 1 mM sodium pyruvate, 0.1 mM nonessential amino acids, 100 U/ml penicillin, 100 µg/ml streptomycin (all from Lonza), along with either 10 mM glucose or 10 mM galactose (Sigma-Aldrich). For mixed lymphocyte cultures, BALB/c CD4$^+$ T cells were purified from adoptively transferred recipients and stimulated (0.3 × 10$^6$ per well) with irradiated B6 spleen cells (1 × 10$^6$ per well). After 48 hr, culture supernatants were tested by ELISA for the production of IFNγ. For anti-CD3/CD28-mediated stimulation of purified T cells, 96-flat-bottomed-well plates were coated overnight at 4°C with different concentrations of anti-CD3 (145–2 C11; BD Biosciences) in PBS. Purified T cells were plated at a concentration of 0.3 × 10$^6$ cells per well and stimulated in the presence of soluble anti-CD28 (37.51; BD Biosciences; 2 µg/ml). For whole spleen cell culture, cells (0.5 × 10$^6$ per well) from the spleen of male recipients that received anti-male CD4$^+$ T cells were incubated in the presence of anti-PD-L1 (10 µg/ml) or control antibodies for 48 hr in 96-round-bottomed well plates. Culture supernatants were then harvested for IFNγ quantification. In some experiments, purified anti-male CD4$^+$ T cells (2 × 10$^5$ per well) were stimulated with bone-marrow-derived dendritic cells (5 × 10$^4$ per well) activated beforehand by overnight LPS treatment (100 ng/ml) (*Buonocore et al., 2003*) and pulsed with different concentrations of male peptides (Eurogentec). Three days later, culture supernatants were collected for IFNγ quantification. For PD-1 engagement experiments, 96-round-bottomed well culture plates were coated overnight at 4°C with anti-CD3 (10 µg/ml) and PD-L1.Fc or control IgG (5 µg/ml) (R and D Systems) in PBS. Purified T cells were plated at a concentration of 0.3 × 10$^6$ per well and stimulated in the presence of 2 µg/ml anti-CD28. 2-Deoxy-D-glucose (2-DG), oligomycin, Etomoxir, or 6-Diazo-5-oxo-L-norleucine (DON) were purchased from Sigma-Aldrich. In experiments assessing T cell proliferation, purified T cells were stained with CFSE, then stimulated in vitro with anti-CD3/CD28 antibodies before CFSE decay was analyzed by flow cytometry.

## Quantification of cytokine production in culture supernatants

Murine cytokine (IFNγ and IL-2) levels were determined in cell-free supernatants using Duoset ELISA (R and D Systems) with detection limits of 20 pg/ml (IFNγ) or 15 pg/ml (IL-2).

## RNA isolation and quantitative PCR

Total RNA from cells was extracted with the RNeasy Mini Kit (QIAGEN). Reverse transcription and real-time PCR reactions were then performed using the Taqman RNA Amplification Kit (one-step procedure) on a LightCycler 480 Real-Time PCR System (Roche). The expression levels of specific mRNA were normalized to the expression of β-actin. β-actin primers: Fw: 5'-TCCTGAGCGCAAG TACTCTGT-3', Rv: 5'-CTGATCCACATCTGCTGGAAG-3'. β-actin probe: 5'-ATCGGTGGCTCCATCC TGGC-3'. Primer and probe mixes for *Glut1*, *Ifng, Fatp* and *Cpt1a* expression were purchased from Roche. For experiments analysing RNA decay, T cells were stimulated in cultures for 6 hr with anti-CD3/CD28. Actinomycin D (Sigma-Aldrich) in DMSO was then added to the culture at a concentration of 10 μM. IFNγ mRNA was quantified by quantitative PCR. Relative expression of mRNA was calculated relative to cultures that received DMSO alone.

## Glucose and lactate measurements in culture supernatants

All supernatants were deproteinized using ultra-centrifugation filter tubes with a 10 kDa cutoff (VWR). Glucose and lactate concentrations were measured using the enzymatic assays commercialized by CMA Microdialysis AB on a CMA600 analyzer (Aurora Borealis) according to manufacturer's instructions.

## OCR and ECAR measurements

Oxygen consumption rates (OCR) and extracellular acidification rates (ECAR) were measured on an XF-96 Extracellular Flux Analyzer (Seahorse Bioscience). Purified CD4$^+$ T cells were suspended in XF base medium (DMEM without glucose) supplemented with 1% heat-inactivated FCS, 10 mM L-glutamine and 10 mM glucose and plated onto Seahorse cell plates ($5 \times 10^5$ cells per well) coated with 200 μg/ml poly-D-lysine (Sigma-Aldricht) to enhance T cell attachment. OCR and ECAR measurements were realized with Agilent Seahorse XF Cell Mito Stress Test Kit (Agilent) according to manufacturer's instruction. Experiments were done with the following assay conditions: 2 min mixture, 2 min wait, and 4 min measurement. Data were normalized for cell numbers by staining with DAPI.

## Intracellular ATP measurement

Intracellular ATP measurements were performed with the ATP bioluminescence assay kit HS II (Roche) according to manufacturer's instructions.

## Flow cytometry

Multi-color flow cytometry was performed on a CyAn ADP LX 9 Color with Summit version 4.3 software (DakoCytomation). For staining, cells were resuspended in PBS with 1% heat-inactivated FCS and were incubated for 10 min with Mouse BD Fc Block™ (BD Biosciences) followed by 30 min in the dark at 4°C with fluorochrome-conjugated anti-CD4 (clone RR4-5), anti-TCRBV6 (clone RR4-7), anti-CD62L (clone MEL14), anti-CD44 (clone IM7), anti-LAG3 (clone C9B7W) and anti-2B4 (clone 2B4) from BD Biosciences, and anti-PD-1 (clone J43) from eBioscience. For the detection of cell surface GLUT1 expression, cells were fixed with formaldehyde 1.5% (VWR), stained with the anti-GLUT1 antibody (1 μg per $10^6$ cells) (Clone EPR3915, Abcam), followed by biotin-labeled goat anti-mouse Ig (dilution 1:50; BD Biosciences) and PerCP-conjugated streptavidin (dilution 1:200; BD Biosciences). To assess cell viability, cells were stained with the fixable viability dye eFluor 780 (DyeFL; 1 μl/$10^6$ cells/ml; eBioscience). Mitochondrial mass was measured by fluorescence levels upon staining with Mitotracker Green (Invitrogen) at 50 nM for 30 min in the dark at 37°C, according to manufacturer's instructions. Mitochondrial ROS measurements were carried out by staining with MitoSOX Red (Invitrogen) at 5 μM for 10 min at 37°C. MitoTracker Red CMXRos stains mitochondria in live cells and its accumulation is dependent upon membrane potential. Cells were incubated for 20 min as indicated in the manufacturer's protocole. For intracytoplasmic immunolabeling, cells were fixed in 0.01% formaldehyde for 10 min, followed by disruption of membrane by Saponin at 0.5% v/v in PBS. Cells were then incubated for 30 min with PE-conjugated anti-mouse IFNγ antibodies (BD Biosciences). For

glucose up-take analysis, purified T cells were incubated with 2-NBDG (Life Technologies) (100 µM) for 30 min. The 2-NBDG uptake reaction was stopped by removing the incubation medium and washing the cells with pre-cold PBS. For Bodipy uptake, cells were incubated with Bodipy FL $C_{16}$ ((4,4-Difluoro-5,7-Dimethyl-4-Bora-3a,4a-Diaza-s-Indacene-3-Hexadecanoic Acid) for 4 hr. Cells were then washed and analyzed by flow cytometry. For 2-NBDG or Bodipy staining, vital dye eFluor 780 was added before analysis to distinguish the viable cell population.

## Two-dimensional gel electrophoresis and mass spectrometry

For protein extraction and two-dimensional gel electrophoresis, purified naive and chronically stimulated $CD4^+$ T cells were lysed in RIPA lysis buffer (Santa Cruz). Proteins were precipitated from the extract supernatant with ice-cold acetone overnight at 20°C. The dried protein pellet was dissolved in the 2DE sample buffer containing 4% CHAPS, 7 M urea, 2 M thiourea, 1% DTT and 1% IPG buffer 3–11 NL (GE Healthcare). The protein concentration was determined using the Bradford protein assay (Sigma-Aldricht). For each 2DE, 90 µg of protein extract was loaded on 18 cm long immobiline strips (pH 3–11 NL) (GE Healthcare) which were first rehydrated overnight at room temperature with 450 µl of rehydration solution (CHAPS 4% (w/v), Urea 7 M, thiourea 2 M, IPG buffer (pH 3–11 NL) 1% (v/v), Destreak reagent (GE Healthcare) 12 µl/mL). Protein extracts were deposited on the strip using cup loading with sample cups for manifold (GE Healthcare), allowing a good separation without any streaking on the basic side of the 2D-gels. Isofocusing was carried out at -20°C for 70 kV-hour with an Ettan IPGphor 2 unit (GE Healthcare). After pH equilibration for 15 min in a solution of Urea 6 M, Tris-HCL 50 mM at pH 8.8, glycerol 30% (v/v) and SDS 2% (w/v) containing 1% of DTT (w/v), the proteins were alkylated for 15 min in the same solution containing 2.5% of iodoacetamide (w/v). The strips were deposited on a 12% SDS-Polyacrylamide gel. Gels were stained overnight with colloidal Coomassie brilliant blue G250.

For qualitative and quantitative image analysis, biological replicates of 2DE were performed for naive (n = 3) and chronically-stimulated (n = 5) $CD4^+$ T cells. Gels (*Fig. 1S*) were scanned with Typhoon (GE Healthcare) and analyzed using SameSpots software v4.1 (Nonlinear Dynamics Ltd). Images were exported for spot detection, matching, normalization and quantification. Spots were quantified on the basis of their normalized volume, that is, the spot volume divided by the total volume of the whole set of gel spots. Spots were scored as variable if they fulfilled the following stringent statistical criteria: ANOVA p≤0.05, fold difference ≥2, q < 0.05 and power >0.8.

For protein identification, major spots, revealed by Coomassie Brilliant Blue (CBB) staining, were subjected to in-gel trypsin digestion according to previously published protocol (*Shevchenko et al., 1996*). The tryptic peptides were separated on a nano-HPLC EASY-nLCII (Thermo Fisher Scientific) coupled to a QTOF (Waters). Peptides were loaded on a 10 cm column with 75 µm inner diameter, packed with 3 µm C18 particles. Reversed-chromatography was performed with a binary buffer system consisting of 0.1% FA (Buffer A) and 95% acetonitril in 0.1% FA (Buffer B) for one-hour gradient run with a flow rate of 300 nl/min. The QTOF was operated in the data dependent mode and the three most abundant isotope patterns with +2 and+3 charges were selected. The raw files were processed using Mascot Daemon platform. The fragmentation spectra were searched against NCBInr with the parent ion mass tolerance set to 100 ppm, and with one miscleavage. Carbamidomethylation of cysteine was set as fixed modification and methionine oxidation and pyroglutamic acid as variable modifications for database searching. Tryptic digests were analyzed on an Ultraflex ToF/ToF mass spectrometer (Bruker, Bremen, Germany). Spectra were processed using FlexAnalysis 2.2 and Biotools 2.2 software (Bruker). MSMS spectra were obtained on the three most abundant peaks of the PMF spectrum. The protein mass list was first searched against the National Center for Biotechnology Information (NCBI) non-redundant protein (nr) database using in-house Mascot software (http://www.matrixscience.com). If the search did not result in identification of a protein, the mass list was searched against the Institute of Genomic Research (TIGR) wheat tentative consensus (TC) database (http://www.tigr.org) or the database of ESTs other than those from human and mouse. The search criteria were monoisotopic mass accuracy <50 ppm, one missed cleavage, and complete carbamidomethylation of cysteine, partial oxidation of methionine and pyroglutamic acid as allowed modifications.

## Western blotting

SDS polyacrylamide gel (SDS–PAGE) and immunoblotting were performed according to standard procedures. Briefly, cells were lysed by RIPA lysis buffer (Santa Cruz, Heidelberg, Germany) on ice. Cell lysates with equal amounts of proteins (15 µg) were separated in 12% SDS–PAGE. Separated proteins were then electrophoretically transferred to a polyvinylidene difluoride membrane (GE Healthcare, Diegem, Belgium), which was subsequently blocked at 4°C for 1 hr with 5% non-fat dry milk in TBST (20 mM Tris, pH 7.6, 137 mM NaCl, 0.1% Tween 20). The blots were then incubated with appropriate dilutions of primary antibodies overnight at 4°C in TBST containing 5% nonfat dry milk. Primary antibodies used for Western blot analysis include rat polyclonal antibodies for p4EBP1, S6, pS6, pEif2 (Abcam, Cambridge, UK) and mouse monoclonal antibody for GAPDH (dilution 1:2000, Meridian Life Science, Memphis, USA). After three washes with TBST, the blots were incubated with horseradish peroxidase-conjugated secondary antibodies either against rat (dilution 1:1000, GE Healthcare) or against mouse (dilution 1:2000, BD Biosciences) in TBST with 5% milk. After several washes with TBST, the blots were incubated at room temperature for 5 min with ECL (Lumigen, Southfield, USA). This was followed by detection with the ChemiDoc XRS (BioRad Laboratories, Temse, Belgium).

## Histology

Liver from male recipients adoptively-transferred with male-specific GFP$^+$$Pdcd1^{+/+}$ CD4$^+$ T cells were fixed in 4% paraformaldehyde, cryoprotected with 30% w/v sucrose in PBS, and frozen in Tissue-Tek OCT compound. Eight-µm cryostat sections were then prepared. Sections were stained with DAPI to label nuclei. The sections were visualized on a Nikon Eclipse 80i microscope and recorded by Nikon Digital Camera DXM1200F. GFP$^+$ cells were counted and the number of cells per mm2 calculated and reported on a graph.

## Statistical analysis

All statistical analyses were performed using Prism 5.0 (GraphPad Software). All p values are two-tailed using Mann-Whitney U test. Since $p > 0.01$ significance can only be reached with experimental group size of minimum five when the Mann Whitney statistical test is applied, experiments were carried out most of the time with groups of minimum five mice or with experimental conditions including five replicated cultures. P values were calculated with lower and upper 95% confidence intervals.

## Acknowledgements

This work is dedicated to the memory of Aimable Rengero Lema. We thank P Horlait and care takers of IMI animal housing facility for their dedicated work. Funding was provided by the Belgian Télévie, Interuniversity Attraction Poles (BELSPO contracts P7/39 and UP7-03), the European Research Council (ERC Starting grant No. 243188 TUMETABO to PS), the *Communauté Française de Belgique* (ARC 09/14–020), the *Fonds National de la Recherche Scientifique* (FRS-FNRS) and the *Fondation Belge contre le Cancer*. PS is a Senior Research Associate and PEP was a Postdoctoral Researcher of the FRS-FNRS at the time of the study. The authors declare no competing financial interests.

# Additional information

## Funding

| Funder | Author |
| --- | --- |
| Televie | Marie Bettonville<br>Stefania d'Aria<br>Kathleen Weatherly |
| Interuniversity Attraction Poles | Pierre Sonveaux<br>Michel Y Braun |
| European Research Council | Pierre Sonveaux |
| ARC from Communaute Francaise de Belgique | Pierre Sonveaux |

| Fonds De La Recherche Scientifique - FNRS | Pierre Sonveaux<br>Michel Y Braun |
| Fondation Belge contre le Cancer | Michel Y Braun |

The funders had no role in study design, data collection and interpretation, or the decision to submit the work for publication.

## Author contributions

Marie Bettonville, Stefania d'Aria, Data curation, Formal analysis, Validation, Investigation, Methodology, Writing—review and editing; Kathleen Weatherly, Jinyu Zhang, Investigation, Methodology; Paolo E Porporato, Conceptualization, Resources, Writing—review and editing; Sabrina Bousbata, Resources, Investigation, Methodology, Writing—review and editing; Pierre Sonveaux, Conceptualization, Resources, Supervision, Funding acquisition, Writing—review and editing; Michel Y Braun, Conceptualization, Resources, Formal analysis, Supervision, Funding acquisition, Validation, Investigation, Visualization, Methodology, Writing—original draft, Project administration, Writing—review and editing

## Author ORCIDs

Stefania d'Aria http://orcid.org/0000-0003-4693-4565
Paolo E Porporato http://orcid.org/0000-0001-8519-1552
Michel Y Braun http://orcid.org/0000-0002-1417-4189

## Ethics

Animal experimentation: This study was performed in strict accordance with the European Union Directive 2010/63/EU on the protection of animals used for scientific purposes. All of the animals were handled according to approved institutional animal care and use committee protocols. The protocol was approved by the Committee on the Ethics of Animal Experiments of the Universite Libre de Bruxelles (Protocol Number: 03-2015).

## Decision letter and Author response

Decision letter https://doi.org/10.7554/eLife.30938.015
Author response https://doi.org/10.7554/eLife.30938.016

# Additional files

## Supplementary files

• Supplementary file 1. 2D-gel identification of proteins differentially expressed by naive or chronic T cells.
DOI: https://doi.org/10.7554/eLife.30938.011

• Supplementary file 2. Two-dimensional gel electrophoresis of proteins from purified naive and chronic CD4 T cells. The figure presents examples of protein separation obtained in this study.
DOI: https://doi.org/10.7554/eLife.30938.012

• Transparent reporting form
DOI: https://doi.org/10.7554/eLife.30938.013

## Data availability

All data generated or analysed during this study are included in the manuscript and supporting files.

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
