## [Decision Letter]

Thank you for submitting your article "Long-term antigen exposure deeply modifies metabolic requirements for T cell function" for consideration by *eLife*. Your article has been reviewed by three peer reviewers, and the evaluation has been overseen by a Reviewing Editor and Wendy Garrett as the Senior Editor. The following individuals involved in review of your submission have agreed to reveal their identity: Lydia Lynch (Reviewer #1).

The reviewers have discussed the reviews with one another and the Reviewing Editor has drafted this decision to help guide your efforts to bring the story to more substantial conclusions.

The manuscript by Bettonville and colleagues addresses an important question in T cell biology, namely the mechanisms that govern the functionality of T cells during chronic antigen exposure. Effector T cells (Teff) under chronic antigen exposure enter a state of non-responsiveness (commonly coined "exhaustion") marked by reduced or absent production of cytokines such as IFNγ. The authors use several adoptive transfer models (i.e. BALB/c CD4^+^ T cell *Rag2^-/-^* B6 allogenic model, anti-male TCR Tg T cells male recipients) to study mechanisms of T cell non-responsiveness, and have focused on metabolic regulation in this paper.

The manuscript is clear and provides some new insight into the metabolic nature of "late" chronically activated CD4^+^ T cells in vivo. Recent work by several groups suggest that glycolysis is a key regulator of IFNγ production in effector T cells. Work by Chang et al. (2013) demonstrated that glycolysis allows for increased translation of IFNγ mRNA. Work by the Gajewski group also established that 2-DG and other forms of glycolytic inhibition could impact IFNγ production by effector T cells. In this work, Bettonville and colleagues propose that chronically activated T cells in vivo are in a state of metabolic non-responsiveness, which may be underscored by reduced mitochondrial activity. However, overall the work is somewhat incremental over recently published results from John Wherry's group characterizing reduced mitochondrial function of exhausted T CD8^+^ cells. The authors describe the metabolic phenotype of chronic CD4^+^ T cells, but provide no mechanistic data to identify what pathways of mitochondrial metabolism chronic CD4 cells are using (beyond glycolysis) to remain functional. In this light, the data are preliminary and additional mechanistic work is needed to make the article suitable for publication in this journal.

The authors also fail to provide insight into why their results contrast with others in the literature. For example, the Wherry Lab have shown that PD1 altered both glycolytic and mitochondrial bioenergetics, based on 8 day analysis as well as some long term measurements (35 days). Engagement of PDL1 results in reduced mitochondrial mass and increased glucose uptake on virus specific T cells. In contrast the authors study shows that blocking PD1 restores function but not glycolysis, and the mitochondrial results contrast also. Based on this, could some differences be due to measuring all PD1+ cells (rather than intermediate vs. hi) and did the authors see any difference seen between PD1 hi and intermediate T cells in this study? Likewise, chronic T cells in their system have low metabolic activity marked by low glycolysis (ECAR), basal OXPHOS (OCR), and low mitochondrial respiratory capacity (SRC), as well as reduced mitochondrial content (Figure 3A-B). This is consistent with recently published work on LCMV chronic infection (Bengsch, Immunity, 2016), although PD-1 knockout cannot restore glycolysis in their system as was suggested in the Bengsch paper, suggesting an alternative mechanism. PD-1-/- T cells in their system still have reduced ECAR/OCR/SRC levels compared to Teff cells (similar to naïve T cells), but can produce IFNγ (Figure 4). The authors claim that these data suggest that engaging glycolysis is not required for IFNγ production by chronic T cells, making their metabolic requirements distinct from Teff cells (Chang et al., 2013). The lack of increased glycolysis in chronic CD4^+^ PD-1-/- T cells is also counter to that observed with CD8^+^ PD-1-/- T cells that have increased glucose uptake and lower mitochondrial mass (Bengsch, 2016).

There are several experimental issues that the authors need to address beyond these issues of contrast with the literature:

1) In Figure 4 the authors show that ECAR of functional *Pdcd1^-/-^* T cells is similar to non-functional *Pdcd1^+/+^* T cells, but that *Pdcd1^-/-^* T cells have substantial OXPHOS. The authors see little effect on the inhibition of IFNγ by 2-DG in these cells. However, one should really test the role of OXPHOS pathways in *Pdcd1^-/-^* T cells. The authors could use DON, Oligo, and Etomoxir as in Figure 3 as reagents to test this.

2) As an extension of point #1 above, the data infer that *Pdcd1^-/-^* T cells use a fuel source or metabolism distinct from *Pdcd1^+/+^* T cells that influences OXPHOS to maintain functionality. Identifying the metabolic pathway(s) involved in this process would distinguish this work substantially from what has been previously published.

3) Regarding the data in Figure 5, it is not clear how ROS affects cytokine production in chronic CD4 cells. Is IFNγ production regulated at the mRNA level or translation, and how does NAC affect this process (i.e. Figure 5F)? The authors can start by measuring IFNγ mRNA levels in control and *Pdcd1^-/-^* T cells after 24h NAC treatment. One mechanistic explanation is that elevated mROS could inactivate mTORC1 (see Li, Cell Signal, 2010), leading to reduced translation. This could be verified using p-S6 phosflow or western blots to measure mTORC1 in their chronic cells. If mTORC1 is reduced, they could use genetic means (i.e. Tsc2 knockdown) to restore mTORC1 activity and see if this reverses the functional defect in their chronic CD4 T cells. Can the authors use NAC or MitoQ/MitoTEMPO in vivo to determine whether limiting ROS can prevent the dysfunctional phenotype? Likewise, ROS has been shown to promote NFAT activation and downstream transcription. Is this axis altered in chronic T cells +/- PD1?

4) What is the phenotype beyond simply PD1 expression in the authors models? PD1 is of course a popular and dominant coinhibitory model but the first marker in a set that defines so-called 'exhausted' T cells. Given that antigen exposure is a dominant driver of T cell exhaustion and previous work in tumor-infiltrating hyporesponsive cd8 T cells (Scharping et al., Immunity 2016, not referenced in this manuscript) suggests that metabolic suppression (glucose uptake and mitochondrial mass) correlates with the degree of exhaustion, it would be important to note what these cells look like in terms of LAG3, TIM3, 2B4, etc. expression.

5) To determine a role for PD1 in the metabolic phenotype, the authors use knockout cells in their system. I would also suggest using PD-L1 blockade in this system, as they did in Figure 1. Likewise, experimental details regarding the results of Figure 4B-C would be important. Is there a source of PD-1 ligation in this system? That may reveal a phenotype for PD-1 potentially, although previous work in tumors suggest PD-1 may be not playing a role in the mitochondrial suppression phenotype.

[Editors' note: further revisions were requested prior to acceptance, as described below.]

Thank you for resubmitting your work entitled "Long-term antigen exposure deeply modifies metabolic requirements for T cell function" for further consideration at *eLife*. Your revised article has been favorably evaluated by Wendy Garrett (Senior Editor), a Reviewing Editor, and two reviewers.

The manuscript has been improved but there are some remaining issues that need to be addressed before acceptance, as outlined below:

In particular, please consider revising the title as suggested, commenting further on the two cited studies in Immunity, and if feasible, adding the mitochondrial data. However, if the latter would substantially delay publication of the paper, some discussion of this issue would be an adequate response in lieu of additional experimental studies.

Reviewer #1:

The authors have provided sufficient additional experiments to answer my comments. I also think they have addressed potential reasons why their results differ from the literature. Thus, I believe that this manuscript provides an important addition to the field, and illustrates that the exhaustion and reinvigoration pathways can be different/more complicated depending on degree of exhaustion.

Reviewer #3:

Generally, the authors addressed many of my initial concerns and this has resulted in an improved manuscript. However, there is still a lack of mechanistic insight in the manuscript: how do the authors suggest these metabolic changes are occurring during chronic stimulation? A lot of this manuscript is profiling WT and KO cells without really asking what is happening when these CD4^+^ T cells are chronically stimulated.

The authors have spent a lot of time and effort refuting parallel but distinct observations in LCMV clone 13 (Bengsch et al., 2016) when the companion paper in that issue, in the tumor immunology field (Scharping et al., Immunity 2016) agrees quite nicely with their findings (decreased mitochondrial mass in exhausted T cells that was not rescued with PD-1 blockade). The authors might want to examine whether expression of some mitochondrial biogenesis genes are altered in their cells and how that might be distinct from LCMV.

---

## [Author Response]

[…] The manuscript is clear and provides some new insight into the metabolic nature of "late" chronically activated CD4^+^ T cells in vivo. Recent work by several groups suggest that glycolysis is a key regulator of IFNγ production in effector T cells. Work by Chang et al. (2013) demonstrated that glycolysis allows for increased translation of IFNγ mRNA. Work by the Gajewski group also established that 2-DG and other forms of glycolytic inhibition could impact IFNγ production by effector T cells. In this work, Bettonville and colleagues propose that chronically activated T cells in vivo are in a state of metabolic non-responsiveness, which may be underscored by reduced mitochondrial activity. However, overall the work is somewhat incremental over recently published results from John Wherry's group characterizing reduced mitochondrial function of exhausted T CD8^+^ cells. The authors describe the metabolic phenotype of chronic CD4^+^ T cells, but provide no mechanistic data to identify what pathways of mitochondrial metabolism chronic CD4 cells are using (beyond glycolysis) to remain functional. In this light, the data are preliminary and additional mechanistic work is needed to make the article suitable for publication in this journal.

We agree with the reviewer's remarks concerning the identity of the metabolic pathway used by chronic T cells in our study to produce IFNγ. In our initial submission to *eLife*, we had shown that, contrary to limiting glycolysis, limiting OXPHOS or FAO inhibited significantly the production of IFNγ by chronic T cells (Figure 4A). This observation led to the hypothesis that fatty acids could support IFNγ production in these T cells. This assumption was also supported by the observation made by Byersdorfer and colleagues (Byersdorfer et al., 2013) that T cells responsible for graft-versus-host disease in mice used FAO to support their effector function. We investigated this possibility by analysing the capacity of chronic T cells to import fatty acids and demonstrated the specific expression of the fatty acid transporter FATP4 as well as the expression of CPT1a responsible for the mitochondrial transport of long chain fatty acids such as palmitoyl-CoA (Figure 4D). Accordingly, chronic T cells were also more sensitive to Bodipy staining, demonstrating a higher capacity to import fatty acids. Finally, we also observed that inhibiting FAO restored the capacity of chronic T cells to consume glucose. However, glucose consumption (with no lactate produced) did not appear to be sufficient for IFNγ synthesis, demonstrating again the incapacity of chronic T cells to mount a glycolytic activity to support IFNγ production. Taken together, our results demonstrate that chronic T cells in our study do use fatty acids as a source of energy for IFNγ. These new observations are presented in the new paragraph added at the end of the subsection “Effector function of chronic CD4^+^ T cells does not require a metabolic shift towards glycolysis”.

We also analyzed fatty acid uptake in *Pdcd1^-/-^* chronic T cells and showed that, though their capacity to import fatty acids from their environment appeared the same as for *Pdcd1^+/+^* cells, *Pdcd1^-/-^* cells expressed higher levels of CPT1a. Thus, the absence of PD-1-mediated regulation of TcR signaling would improve chronic T cell's capacity to oxidize fatty acids. This is presented in the aforementioned paragraph.

The authors also fail to provide insight into why their results contrast with others in the literature. For example, the Wherry Lab have shown that PD1 altered both glycolytic and mitochondrial bioenergetics, based on 8 day analysis as well as some long term measurements (35 days). Engagement of PDL1 results in reduced mitochondrial mass and increased glucose uptake on virus specific T cells. In contrast the authors study shows that blocking PD1 restores function but not glycolysis, and the mitochondrial results contrast also. Based on this, could some differences be due to measuring all PD1+ cells (rather than intermediate vs. hi) and did the authors see any difference seen between PD1 hi and intermediate T cells in this study? Likewise, chronic T cells in their system have low metabolic activity marked by low glycolysis (ECAR), basal OXPHOS (OCR), and low mitochondrial respiratory capacity (SRC), as well as reduced mitochondrial content (Figure 3A-B). This is consistent with recently published work on LCMV chronic infection (Bengsch, Immunity, 2016), although PD-1 knockout cannot restore glycolysis in their system as was suggested in the Bengsch paper, suggesting an alternative mechanism. PD-1-/- T cells in their system still have reduced ECAR/OCR/SRC levels compared to Teff cells (similar to naïve T cells), but can produce IFNγ (Figure 4). The authors claim that these data suggest that engaging glycolysis is not required for IFNγ production by chronic T cells, making their metabolic requirements distinct from Teff cells (Chang et al., 2013). The lack of increased glycolysis in chronic CD4^+^ PD-1-/- T cells is also counter to that observed with CD8^+^ PD-1-/- T cells that have increased glucose uptake and lower mitochondrial mass (Bengsch, 2016).

As mentioned by the reviewer, our study contrasts with what has previously been published in the literature about the metabolism of exhausted T cells and adds some important new facts for the understanding of the biology of T cells under chronic antigen stimulation. Our data fit with the general concept that T cell unresponsiveness is a progressive state that evolves from partial to full unresponsiveness. We show that 1) chronic T cells exposed to long-term antigen stimulation express higher levels of inhibitory receptors than early chronic T cells; 2) contrary to early chronic T cells, late chronic T cells have lost some of their function (IL-2 production and proliferation) that cannot be restored. Our data add the observation that early chronic antigen exposure does not alter T cell capacity to monopolize glycolysis for effector function whereas after late chronic exposure T cells are unable to engage glycolysis. The main difference with the recent published work on LCMV chronic infection by Bengsch and co-workers, where glycolysis is engaged for early and late chronic T cell function, is that in our adoptive transfer model T cells are exposed to chronic antigen stimulation in the absence of inflammatory signals. Thus, chronic antigen recognition by T cells would be carried out without CD28 co-stimulation. Since we know that sustained glycolytic function in T cells requires CD28 signaling (Menk et al., 2018) and that rendering T cells anergic through TcR stimulation alone not only prevents upregulation of glycolytic metabolism, but also prevents it from being upregulated in future stimulations, even those delivered with costimulation (Zheng et al., 2009), one could expect that chronic antigen stimulation in the absence of costimulation would lead to the maintenance of a state where glycolysis could not be engaged for T cell function. We have added several paragraphs in the Discussion section of the manuscript presenting this new concept.

Finally, it should be pointed out that PD1-/- T cells in our system have reduced ECAR/OCR/SRC (and glycolysis) levels compared to Teff cells, but can produce IFNγ (Figure 5). However, unlike LCMV-specific exhausted T cells, their proliferation is not restored after stimulation in the absence of PD-L1-PD1 interactions (Figure 2F), leaving most of their oxidative energy supply for the synthesis of IFNγ alone. This could explain why chronic T cells are still capable to produce IFNγ despite their deficit in metabolic pathways.

There are several experimental issues that the authors need to address beyond these issues of contrast with the literature:1) In Figure 4 the authors show that ECAR of functional Pdcd1^-/-^ T cells is similar to non-functional Pdcd1^+/+^ T cells, but that Pdcd1^-/-^ T cells have substantial OXPHOS. The authors see little effect on the inhibition of IFNγ by 2-DG in these cells. However, one should really test the role of OXPHOS pathways in Pdcd1^-/-^ T cells. The authors could use DON, Oligo, and Etomoxir as in Figure 3 as reagents to test this.

We have performed the experiments requested by the reviewer and tested the capacity of DON, Oligo and Etomoxir to inhibit the production of IFNγ by *Pdcd1^-/-^* after anti-CD3/CD28 stimulation. As seen in the new Figure 5E, *Pdcd1^-/-^* T cells exhibited a similar sensitivity to inhibition by Etomoxir and Oligomycin as *Pdcd1^+^* T cells, whereas they were not sensitive to 2-DG and DON inhibition. Thus, like *Pdcd1^+/+^* T cells, *Pdcd1^-/-^* T cells appear to use FAO and OXPHOS for the production of IFNγ.

2) As an extension of point #1 above, the data infer that Pdcd1^-/-^ T cells use a fuel source or metabolism distinct from Pdcd1^+/+^ T cells that influences OXPHOS to maintain functionality. Identifying the metabolic pathway(s) involved in this process would distinguish this work substantially from what has been previously published.

We agree with the reviewer. As mentioned above in our reply to the reviewers' first remark, we have accumulated evidence demonstrating that chronic T cells in our system use FAO to support IFNγ production. These are presented in Figure 4.

3) Regarding the data in Figure 5, it is not clear how ROS affects cytokine production in chronic CD4 cells. Is IFNγ production regulated at the mRNA level or translation, and how does NAC affect this process (i.e. Figure 5F)? The authors can start by measuring IFNγ mRNA levels in control and Pdcd1^-/-^ T cells after 24h NAC treatment. One mechanistic explanation is that elevated mROS could inactivate mTORC1 (see Li, Cell Signal, 2010), leading to reduced translation. This could be verified using p-S6 phosflow or western blots to measure mTORC1 in their chronic cells. If mTORC1 is reduced, they could use genetic means (i.e. Tsc2 knockdown) to restore mTORC1 activity and see if this reverses the functional defect in their chronic CD4 T cells. Can the authors use NAC or MitoQ/MitoTEMPO in vivo to determine whether limiting ROS can prevent the dysfunctional phenotype? Likewise, ROS has been shown to promote NFAT activation and downstream transcription. Is this axis altered in chronic T cells +/- PD1?

We have carried out new experiments to determine whether regulation of IFNγ secretion by chronic T cells is at the level of transcription or translation. First, we analyzed the phosphorylation of mTORC1 targets involved in protein translation. As seen in Figure 7, low ROSproducer *Pdcd1^+/+^* T cells and high ROS-producer *Pdcd1^-/-^* T cells had the same capacity to phosphorylate mTORC1 targets involved in protein translation, namely 4EBP and S6, after activation. This supported the conclusion that the presence of more abundant ROS does not affect protein translation in chronic T cells. We have also analyzed, as requested by the reviewers, the production of IFNγ mRNA after short stimulation in chronic T cells. As seen in Figure 7D, Pdcd1-/- T cells had consistently more IFNγ mRNA after activation than *Pdcd1^+/+^* T cells. This was surprising since at the protein level, *Pdcd1^+/+^* chronic T cells appeared to produce more IFNγ that *Pdcd1^-/-^* (Figure 7A and B). Taken together these results suggested that mRNA produced in High ROS producer *Pdcd1^-/-^* might be less stable than in *Pdcd1^+/+^* T cells. ROS are a major source of damage to cellular components and RNA damage has been reported to increase during oxidative stress. (Li et al. IUBMB Life 2006 58:582). Taken together, our data support the notion that, in chronic CD4^+^ T cells, PD-1 increases T cell activation threshold to prevent excessive RNA oxidation and decay, consequently preserving the capacity of T cells to produce inflammatory cytokines such as IFNγ (see Discussion, fourth paragraph).

4) What is the phenotype beyond simply PD1 expression in the authors models? PD1 is of course a popular and dominant coinhibitory model but the first marker in a set that defines so-called 'exhausted' T cells. Given that antigen exposure is a dominant driver of T cell exhaustion and previous work in tumor-infiltrating hyporesponsive cd8 T cells (Scharping et al., Immunity 2016, not referenced in this manuscript) suggests that metabolic suppression (glucose uptake and mitochondrial mass) correlates with the degree of exhaustion, it would be important to note what these cells look like in terms of LAG3, TIM3, 2B4, etc. expression.

We have compared the expression of PD1, LAG3 and 2B4 in T cells exposed for 8 days or 21 days to chronic male antigen stimulation. Our results show that day 21 T cells express higher levels of PD1, LAG3 and 2B4 coinhibitory receptors than day 7 T cells. This observation supports the notion that day 21 T cells are in a deeper stage of "exhaustion" than day 7 T cells. The description of these data and conclusion can be found in the second paragraph of the subsection “Effector function of chronic CD4^+^ T cells does not require a metabolic shift towards glycolysis” and in Figure 3I.

5) To determine a role for PD1 in the metabolic phenotype, the authors use knockout cells in their system. I would also suggest using PD-L1 blockade in this system, as they did in Figure 1. Likewise, experimental details regarding the results of Figure 4B-C would be important. Is there a source of PD-1 ligation in this system? That may reveal a phenotype for PD-1 potentially, although previous work in tumors suggest PD-1 may be not playing a role in the mitochondrial suppression phenotype.

Since analyses of the metabolic phenotype (with inhibitors) are carried out with purified *Pdcd1^+/+^* T cells, PD1 molecules present at T cell surface are not engaged by ligands expressed by APC. Thus, in our experiments, T cell metabolic profile is determine in the absence of PD1 inhibition of TcR signalling. Since it has been proposed that PD-L1 can also engaged CD80 in T-T cell contacts and that this interaction can suppress T cell activity, we also analyzed the effect of PDL1 neutralisation on the activity of purified *Pdcd1^+/+^* T cells. As show in Figures 2G-H, neutralisation of PD-L1 by antibodies did not modify the capacity of chronic T cells to produce Ifnγ, demonstrating that regulation of T cell activity by PD-L1-CD80 interaction does not seem to play a role in this system.

[Editors' note: further revisions were requested prior to acceptance, as described below.]

The manuscript has been improved but there are some remaining issues that need to be addressed before acceptance, as outlined below:In particular, please consider revising the title as suggested, commenting further on the two cited studies in Immunity, and if feasible, adding the mitochondrial data. However, if the latter would substantially delay publication of the paper, some discussion of this issue would be an adequate response in lieu of additional experimental studies.Reviewer #3:Generally, the authors addressed many of my initial concerns and this has resulted in an improved manuscript. However, there is still a lack of mechanistic insight in the manuscript: how do the authors suggest these metabolic changes are occurring during chronic stimulation? A lot of this manuscript is profiling WT and KO cells without really asking what is happening when these CD4^+^ T cells are chronically stimulated.

We have added a paragraph on what we believe could be the natural history of chronically stimulated CD4^+^ T cells as suggested by the reviewer (Discussion, last paragraph).

The authors have spent a lot of time and effort refuting parallel but distinct observations in LCMV clone 13 (Bengsch et al., 2016) when the companion paper in that issue, in the tumor immunology field (Scharping et al., Immunity 2016) agrees quite nicely with their findings (decreased mitochondrial mass in exhausted T cells that was not rescued with PD-1 blockade). The authors might want to examine whether expression of some mitochondrial biogenesis genes are altered in their cells and how that might be distinct from LCMV.

We have added a discussion point (Discussion, fourth paragraph) on mitochondria biogenesis in chronic CD4^+^ T cells and, based on what we observed, we think PD-1 does not control mitochondria biogenesis through PGC1a as in the early exhaustion of LCMV-specific T cells.